# Robust Unlearnable Examples: Protecting Data Against Adversarial Learning

**Shaopeng Fu[1], Fengxiang He[1], Yang Liu[2], Li Shen[1] & Dacheng Tao[1]**
[1]JD Explore Academy, JD.com Inc, China
[2]Institute for AI Industry Research, Tsinghua University, China
`{shaopengfu15, fengxiang.f.he, mathshenli, dacheng.tao}@gmail.com`
`liuy03@air.tsinghua.edu.cn`

## ABSTRACT

The tremendous amount of accessible data in cyberspace face the risk of being unauthorized used for training deep learning models. To address this concern, methods are proposed to make data unlearnable for deep learning models by adding a type of error-minimizing noise. However, such conferred unlearnability is found fragile to adversarial training. In this paper, we design new methods to generate robust unlearnable examples that are protected from adversarial training. We first find that the vanilla error-minimizing noise, which suppresses the informative knowledge of data via minimizing the corresponding training loss, could not effectively minimize the adversarial training loss. This explains the vulnerability of error-minimizing noise in adversarial training. Based on the observation, robust error-minimizing noise is then introduced to reduce the adversarial training loss. Experiments show that the unlearnability brought by robust error-minimizing noise can effectively protect data from adversarial training in various scenarios. The code is available at `https://github.com/fshp971/robust-unlearnable-examples`.

## 1 INTRODUCTION

Recent advances in deep learning largely rely on various large-scale datasets, which are mainly built upon public data collected from various online resources such as Flickr, Google Street View, and search engines (Lin et al., 2014; Torralba et al., 2008; Netzer et al., 2011). However, the process of data collection might be unauthorized, leading to the risk of misusing personal data for training deep learning models (Zhang & Tao, 2020; He & Tao, 2020). For example, a recent work demonstrates that individual information such as name or email could be leaked from a pre-trained GPT-2 model (Carlini et al., 2020). Another investigation reveals that a company has been continuously collecting facial images from the internet for training commercial facial recognition models (Hill, 2020).

To prevent the unauthorized use of the released data, recent studies suggest poisoning data with imperceptible noise such that the performances of models trained on the modified data be significantly downgraded (Fowl et al., 2021b;a; Yuan & Wu, 2021; Huang et al., 2021). This work focuses on that by Huang et al. (2021). Specifically, they introduce *unlearnable examples* that are crafted from clean data via adding a type of imperceptible *error-minimizing noise* and show impressive protection ability. The error-minimizing noise is designed based on the intuition that an example of higher training loss may contain more knowledge to be learned. Thereby, the noise protects data from being learned via minimizing the corresponding loss to suppress the informative knowledge of data.

However, such an unlearnability conferred by error-minimizing noise is found vulnerable to adversarial training (Huang et al., 2021; Tao et al., 2021), a standard approach towards training robust deep learning models (Madry et al., 2018). In adversarial training, a model trained on the unlearnable data can achieve the same performance as that trained on the clean data. Similar problems are also found in other poisoning-based data protection methods (Fowl et al., 2021b; Yuan & Wu, 2021) (see Fig. 1). These findings bring great challenges to poisoning-based data protection, and further raises the following question: *Can we still make data unlearnable by adding imperceptible noise even under the adversarial training?*

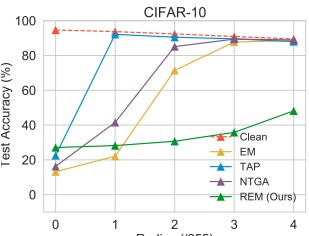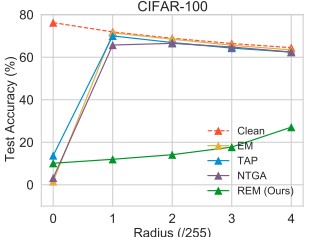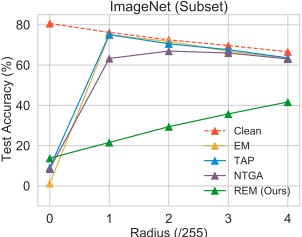

Figure 1: We conduct adversarial training on data protected by different types of noise with varied adversarial perturbation radius $\rho_a$. EM denotes error-minimizing noise, TAP denotes targeted adversarial poisoning noise, NTGA denotes neural tangent generalization attack noise, and REM denotes robust error-minimizing noise. The curves of test accuracy vs. radius $\rho_a$ are plotted. The results show that as the radius $\rho_a$ increases, (1) the protection brought by EM, NTGA and TAP gradually become invalid, and (2) the proposed REM can still protect data against adversarial learners.

In this paper, we give an affirmative answer that imperceptible noise can indeed stop deep learning models from learning high-level knowledge in an adversarial manner. We first analyze the reason why the unlearnability conferred by error-minimizing noise fails under adversarial training. Adversarial training improves the robustness of models via training them on adversarial examples. However, adversarial examples usually correspond to higher loss, which according to Huang et al. (2021) may contain more knowledge to be learned. Though the error-minimizing noise is designed to minimize the training loss, adversarial examples crafted from unlearnable data could still significantly increase the loss scale (see Fig. 2). Therefore, adversarial training can break the protection brought by error-minimizing noise and still learn knowledge from these data.

Based on the aforementioned analysis, we further deduce that the invalidation of unlearnable examples in adversarial training might be attributed to the error-minimizing noise could not effectively reduce the adversarial training loss. To this end, a new type of noise, named *robust error-minimizing noise*, is designed to protect data against adversarial learners via minimizing the adversarial training loss. Inspired by the strategy of adversarial training and the generation of error-minimizing noise, a min-min-max optimization is designed to train the robust error-minimizing noise generator. Examples crafted by adding robust error-minimizing noise are called *robust unlearnable examples*.

In summary, our work has three main contributions: (1) We present robust error-minimizing noise, which to the best of our knowledge, is the first and currently the only method that can prevent data from being learned in adversarial training. (2) We propose to solve a min-min-max optimization problem to effectively generate robust error-minimizing noise. (3) We empirically verify the effectiveness of robust error-minimizing noise under adversarial training.

## 2 RELATED WORKS

**Adversarial attacks.** Adversarial examples are carefully crafted from normal data and can fool machine learning models to behave unexpectedly. It has been found that machine learning models are vulnerable to adversarial examples (Szegedy et al., 2013; Goodfellow et al., 2015; Carlini & Wagner, 2017). Though minor data transformations can mitigate the adversaries of adversarial examples (Lu et al., 2017; Luo et al., 2015), some studies have shown that adversarial examples can further be made resistant to these transformations (Evtimov et al., 2017; Eykholt et al., 2018; Wu et al., 2020b; Athalye et al., 2018; Liu et al., 2019; 2020a). This makes adversarial attacks realistic threatens.

To tackle adversarial attacks, adversarial training is proposed to improve the robustness of machine learning models against adversarial examples. Similar to GANs (Goodfellow et al., 2014; Wang et al., 2017; 2019), a standard adversarial training algorithm aims to solve a minimax problem that minimizes the loss function on *most adversarial examples* (Madry et al., 2018). Other advances include TRADE (Zhang et al., 2019), FAT (Zhang et al., 2020), GAIRAT (Zhang et al., 2021), (Song et al., 2020), Jiang et al. (2021), Stutz et al. (2020), Singla & Feizi (2020), Wu et al. (2020a), and Wu et al. (2021). Tang et al. (2021) propose an adversarial robustness benchmark regarding architecture design and training techniques. Some works also attempt to interpret how machine learning models gain robustness (Ilyas et al., 2019; Zhang & Zhu, 2019).

**Poisoning attacks.** Poisoning attacks aim to manipulate the performance of a machine learning model via injecting malicious poisoned examples into its training set (Biggio et al., 2012; Koh & Liang, 2017; Shafahi et al., 2018; Liu et al., 2020b; Jagielski et al., 2018; Yang et al., 2017; Steinhardt et al., 2017). Though poisoned examples usually appear different from the clear ones (Biggio et al., 2012; Yang et al., 2017), however, recent approaches show that poisoned examples can also be crafted imperceptible to their corresponding origins (Koh & Liang, 2017; Shafahi et al., 2018). A special case of poisoning attacks is the backdoor attack. A model trained on backdoored data may perform well on normal data but wrongly behave on data that contain trigger patterns (Chen et al., 2017; Nguyen & Tran, 2020; 2021; Li et al., 2021; Weng et al., 2020). Compared to other poisoning attacks, backdoor attack is more covert and thus more threatening (Gu et al., 2017).

Recent works suggest employ data poisoning to prevent unauthorized model training. Huang et al. (2021) modify data with error-minimizing noise to stop deep models learning knowledge from the modified data. Yuan & Wu (2021) generate protective noise for data upon an ensemble of neural networks modeled with neural tangent kernels (Jacot et al., 2018), which thus enjoys strong transferability. Fowl et al. (2021a) and Fowl et al. (2021b) employ gradient alignment (Geiping et al., 2020; 2021) and PGD method (Madry et al., 2018) to generate adversarial examples as poisoned data to downgrade the performance of models trained on them. However, Tao et al. (2021) find that existing poisoning-based protection can be easily broken by adversarial training.

## 3 PRELIMINARIES

Suppose $\mathcal{D} = \{(x_1, y_1), \cdots, (x_n, y_n)\}$ is a dataset consists of $n$ samples, where $x_i \in \mathcal{X}$ is the feature of the $i$-th sample and $y_i \in \mathcal{Y}$ is the corresponding label. A parameterized machine learning model is $f_\theta : \mathcal{X} \to \mathcal{Y}$, where $\theta \in \Theta$ is the model parameter. Suppose $\ell : \mathcal{Y} \times \mathcal{Y} \to [0, 1]$ is a loss function. Then, the empirical risk minimization (ERM) aims to approach the optimal model by solving the following optimization problem,

$$\min_\theta \frac{1}{n} \sum_{i=1}^{n} \ell(f_\theta(x_i), y_i). \tag{1}$$

**Adversarial training** (Madry et al., 2018) improves the robustness of models against adversarial attacks via training them on adversarial examples. A standard adversarial training aims to solve the following min-max problem,

$$\min_\theta \frac{1}{n} \sum_{i=1}^{n} \max_{\|\delta_i\| \le \rho_a} \ell(f_\theta(x_i + \delta_i), y_i), \tag{2}$$

where $\rho_a > 0$ is the adversarial perturbation radius and $(x_i + \delta_i)$ is the most adversarial example within the ball sphere centered at $x_i$ with radius $\rho_a$. Usually, a larger radius $\rho_a$ implies stronger adversarial robustness of the trained model.

**Unlearnable examples** (Huang et al., 2021) is a type of data that deep learning models could not effectively learn informative knowledge from them. Models trained on unlearnable examples could only achieve severely low performance. The generation of unlearnable examples contains two steps. Firstly, train an error-minimizing noise generator $f'_\theta$ via solving the following optimization problem,

$$\min_\theta \frac{1}{n} \sum_{i=1}^{n} \min_{\|\delta_i\| \le \rho_u} \ell(f'_\theta(x_i + \delta_i), y_i), \tag{3}$$

where $\rho_u$ is the defensive perturbation radius that forces the generated error-minimizing noise to be imperceptible. Then, an unlearnable example $(x', y)$ is crafted via adding error-minimizing noise generated by the trained noise generator $f'_\theta$ to its clean counterpart $(x, y)$, where $x' = x + \arg\min_{\|\delta\| \le \rho_u} \ell(f'_\theta(x + \delta), y)$. The intuition behind this is, a smaller loss may imply less knowledge that could be learned from an example. Thus, error-minimizing noise aims to make data unlearnable via reducing the corresponding training loss. It is worth noting that once the unlearnable data is released publicly, the data defender could not modify the data any further.

**Projected gradient descent (PGD)** (Madry et al., 2018) is a standard approach for solving the inner maximization and minimization problems in Eqs. (2) and (3). It performs iterative projection

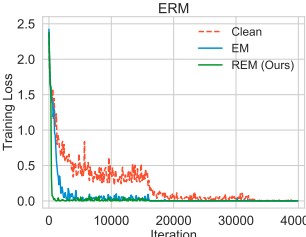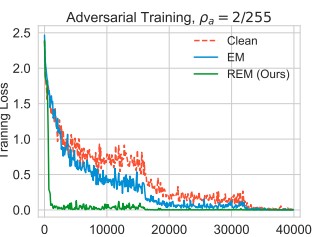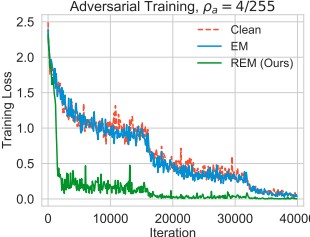

Figure 2: The training loss curves of ERM training and adversarial training on CIFAR-10. Lower training losses suggest stronger unlearnability of data. The results show that: (1) error-minimizing noise could not reduce the training loss as effectively as that in ERM training; (2) robust error-minimizing noise can preserve the training loss in significantly low levels across various learning scenarios. These observations suggest that robust error-minimizing noise is more favorable in preventing data from being learned via adversarial training.

updates to search for the optimal perturbation as follows,

$$\delta^{(k)} = \prod_{\|\delta\| \leq \rho} \left[ \delta^{(k-1)} + c \cdot \alpha \cdot \text{sign} \left( \frac{\partial}{\partial \delta} l(f_\theta(x + \delta^{(k-1)}), y) \right) \right],$$

where $k$ is the current iteration step ($K$ steps at all), $\delta^{(k)}$ is the perturbation found in the $k$-th iteration, $c \in \{-1, 1\}$ is a factor for controlling the gradient direction, $\alpha$ is the step size, and $\prod_{\|\delta\| \leq \rho}$ means the projection is calculated in the ball sphere $\{\delta : \|\delta\| \leq \rho\}$. The final output perturbation is $\delta^{(K)}$. Throughout this paper, the coefficient $c$ is set as 1 when solving maximization problems and $-1$ when solving minimization problems.

## 4 ROBUST UNLEARNABLE EXAMPLES

In this section, we first illustrate the difficulty of reducing adversarial training loss with error-minimizing noise, and then introduce robust unlearnable examples to tackle the raised challenge.

### 4.1 ERROR-MINIMIZING NOISE IN ADVERSARIAL TRAINING

Error-minimizing noise prevents unauthorized model training via reducing the corresponding loss to suppress the learnable knowledge of data. However, the goal of reducing training loss may be at odds with that of adversarial training. Recall Eq. (2), adversarial examples usually correspond to higher training losses than that of the clean data, which according to Huang et al. (2021) may contain more knowledge to be learned. Even if error-minimizing noise presents, crafting adversarial examples from unlearnable data could still significantly increase the training loss.

To better illustrate this phenomenon, we conduct adversarial training on clean data and unlearnable data with different adversarial perturbation radius $\rho_a$. The training losses along the training process are collected and plotted in Fig. 2. From the figure, it can be found that error-minimizing noise could not reduce the training loss as effectively as that in ERM training. Furthermore, as the adversarial perturbation radius $\rho_a$ increases, the adversarial training loss curves on unlearnable data eventually coincide with that on clean data. These observations suggest that adversarial training can effectively recover the knowledge of data protected by error-minimizing noise, which makes the conferred unlearnability invalid.

### 4.2 ROBUST ERROR-MINIMIZING NOISE

Motivated by the empirical study in Section 4.1, we propose a type of imperceptible noise, named robust error-minimizing noise, to protect data against adversarial learners. Robust error-minimizing noise is designed to minimize the adversarial training loss during adversarial training, which intuitively can barrier the knowledge learning process in adversarial training. Examples crafted via adding robust error-minimizing noise are named robust unlearnable examples.

Similar to that in Huang et al. (2021), the generation of robust error-minimizing noise also needs to first train a noise generator $f'_\theta$. Inspired by the adversarial training loss in Eq. (2) and the objective function for training error-minimizing noise generator in Eq. (3), a min-min-max optimization process is designed to train the robust error-minimizing noise generator $f'_\theta$ as follows,

$$\min_\theta \frac{1}{n} \sum_{i=1}^n \min_{\|\delta_i^u\| \le \rho_u} \max_{\|\delta_i^a\| \le \rho_a} \ell(f'_\theta(x_i + \delta_i^u + \delta_i^a), y_i), \tag{4}$$

where the defensive perturbation radius $\rho_u$ forces the generated noise to be imperceptible, and the adversarial perturbation radius $\rho_a$ controls the protection level of the noise against adversarial training. The idea behind Eq. (4) is, we aim to find a "safer" defensive perturbation $\delta_i^{u*}$ such that the adversarial example crafted from the protected sample point $(x_i + \delta_i^{u*})$ would not increase the training loss too much. As a result, the adversarial learner could not extract much knowledge from the data protected by this safer defensive perturbation.

For the choices of the two perturbation radii $\rho_u$ and $\rho_a$, we notice that when $\rho_u \le \rho_a$, the following inequality holds for each summation term in Eq. (4),

$$\min_{\|\delta_i^u\| \le \rho_u} \max_{\|\delta_i^a\| \le \rho_a} \ell(f'_\theta(x_i + \delta_i^u + \delta_i^a), y_i) \ge \ell(f'_\theta(x_i), y_i).$$

The above inequality suggests that when $\rho_u \le \rho_a$, the generated defensive noise $\delta_i^u$ could not suppress any learnable knowledge of data even in ERM training. Thereby, to help the generated noise gain remarkable protection ability, the radius $\rho_u$ should be set larger than $\rho_a$.

## 4.3 Enhancing the Stability of the Noise

However, the unlearnability conferred by the noise generated via Eq. (4) is found to be fragile to minor data transformation. For example, standard data augmentation (Shorten & Khoshgoftaar, 2019) can easily make the conferred unlearnability invalid (see Section 5.2). To this end, we adopt the *expectation over transformation* technique (EOT; Athalye et al. 2018) into the generation of robust error-minimizing noise. EOT is a stability-enhancing technique that was first proposed for adversarial examples (Evtimov et al., 2017; Eykholt et al., 2018; Athalye et al., 2018). Since adversarial examples and robust unlearnable examples are similar to some extent, it is likely that EOT can also help to improve the stability of robust error-minimizing noise.

Suppose $T$ is a given distribution over a set of some transformation functions $\{t : \mathcal{X} \to \mathcal{X}\}$. Then, the objective function for training robust error-minimizing noise generator with EOT is adapted from Eq. (4) as follows,

$$\min_\theta \frac{1}{n} \sum_{i=1}^n \min_{\|\delta_i^u\| \le \rho_u} \mathbb{E}_{t \sim T} \max_{\|\delta_i^a\| \le \rho_a} \ell(f'_\theta(t(x_i + \delta_i^u) + \delta_i^a), y_i). \tag{5}$$

Eq. (5) suggests that, when searching for the defensive perturbation $\delta_i^u$, one needs to minimize several adversarial losses of a set of the transformed examples rather than only minimize the adversarial losses of single examples. After finishing training the noise generator via Eq. (5), the robust unlearnable example $(x', y)$ for a given data point $(x, y)$ is generated as

$$x' = x + \arg\min_{\|\delta^u\| \le \rho_u} \mathbb{E}_{t \sim T} \max_{\|\delta^a\| \le \rho_a} \ell(f'_\theta(t(x + \delta^u) + \delta^a), y).$$

## 4.4 Efficiently Training the Noise Generator

To train the robust error-minimizing noise generator with Eq. (5), we employ PGD to solve the inner minimization and maximization problems. The main challenge of this approach is the gradient calculation of the inner maximization function.

Specifically, the inner maximization function in Eq. (5) usually does not have an analytical solution. As a result, its gradient calculation could not be directly handled by modern Autograd systems such as PyTorch and TensorFlow. Toward this end, we approximate the gradient via first calculating the optimal perturbation $\delta_i^{a*} = \arg\max_{\|\delta_i^a\| \le \rho_a} \ell(f'_\theta(t(x_i + \delta_i^u) + \delta_i^a), y_i)$ with PGD, and then approximating the gradient of the maximization function by $\frac{\partial}{\partial \delta_i^u} \ell(f'_\theta(t(x_i + \delta_i^u) + \delta_i^{a*}), y_i)$.

---

**Algorithm 1** Training robust error-minimizing noise generator with Eq. (5)

---

**Input:** Training data set $\mathcal{D}$, training iteration $M$,
  1: PGD parameters $\rho_u$, $\alpha_u$ and $K_u$ for solving the minimization problem,
  2: PGD parameters $\rho_a$, $\alpha_a$ and $K_a$ for solving the maximization problem,
  3: The data transformation distribution $T$,
  4: the number of sampling time $J$ when approximating the gradient of the expectation function.
**Output:** Robust error-minimizing noise generator $f'_\theta$.
  5: Initialize source model parameter $\theta$.
  6: **for** $i$ **in** $1, \cdots, M$ **do**
  7:     Sample a minibatch $(x, y) \sim \mathcal{D}$.
  8:     Initialize $\delta^u$.
  9:     **for** $k$ **in** $1, \cdots, K_u$ **do**
 10:         **for** $j$ **in** $1, \cdots, J$ **do**
 11:             Sample a transformation function $t_j \sim T$.
 12:             $\delta_j^a \leftarrow \text{PGD}(t_j(x + \delta^u), y, f'_\theta, \rho_a, \alpha_a, K_a)$          $\triangleright$ Finding adversarial perturbation.
 13:         **end for**
 14:         $g_k \leftarrow \frac{1}{J} \sum_{j=1}^{J} \frac{\partial}{\partial \delta^u} \ell(f'_\theta(t_j(x + \delta^u) + \delta_j^a), y)$
 15:         $\delta^u \leftarrow \prod_{\|\delta\| \leq \rho_u} (\delta^u - \alpha_u \cdot \text{sign}(g_k))$
 16:     **end for**
 17:     Sample a transformation function $t \sim T$.
 18:     $\delta^a \leftarrow \text{PGD}(t(x + \delta^u), y, f'_\theta, \rho_a, \alpha_a, K_a)$
 19:     Update source model parameter $\theta$ based on minibatch $(t(x + \delta^u) + \delta^a, y)$.
 20: **end for**
 21: **return** $f'_\theta$

---

Therefore, when searching for the optimal defensive perturbation $\delta_i^{u*}$ with PGD, one can follow Athalye et al. (2018) to approximate the gradient of the expectation of transformation via first sampling several transformation functions and then averaging the gradients of the corresponding inner maximization function. The gradients of the inner maximization function are approximated via the aforementioned gradient approximation method. Finally, the overall procedures for solving Eq. (5) are presented as Algorithm 1. The effect of the robust error-minimizing noise generated based on Algorithm 1 is illustrated in Fig. 2, which verifies the ability of error-minimizing noise of suppressing training loss in adversarial training.

## 5  Experiments

In this section, we conduct comprehensive experiments to verify the effectiveness of the robust error-minimizing noise in preventing data being learned by adversarial learners.

### 5.1  Experiment Setup

**Datasets.** Three benchmark computer vision datasets, CIFAR-10, CIFAR-100 (Krizhevsky et al., 2009), and an ImageNet subset (consists of the first 100 classes) (Russakovsky et al., 2015), are used in our experiments. The data augmentation technique (Shorten & Khoshgoftaar, 2019) is adopted in every experiment. For the detailed settings of the data augmentation, please see Appendix A.2.

**Robust error-minimizing noise generation.** Following Huang et al. (2021), we employ ResNet-18 (He et al., 2016) as the source model $f'_\theta$ for training robust error-minimizing noise generator with Eq. (5). The $L_\infty$-bounded noises $\|\delta_u\|_\infty \leq \rho_u$ and $\|\delta_a\|_\infty \leq \rho_a$ are adopted in our experiments. The detailed training settings are presented in Appendix A.3.1.

**Baseline methods.** The proposed robust error-minimizing noise **(REM)** is compared with three state-of-the-art data protection noises, the error-minimizing noise **(EM)** (Huang et al., 2021), the targeted adversarial poisoning noise **(TAP)** (Fowl et al., 2021b), and the neural tangent generalization attack noise **(NTGA)** (Yuan & Wu, 2021). See Appendix A.3.2 for the detailed generation procedures of every type of noise.

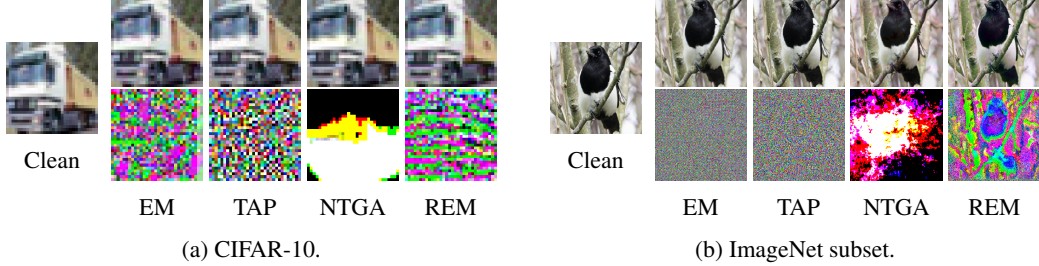

Clean    EM   TAP   NTGA   REM      Clean    EM   TAP   NTGA   REM

(a) CIFAR-10.            (b) ImageNet subset.

Figure 3: Visualization results of different types of defensive noise as well as the correspondingly crafted examples. EM denotes error-minimizing noise, TAP denotes targeted adversarial poisoning noise, NTGA denotes neural tangent generalization attack noise, and REM denotes robust error-minimizing noise.

Table 1: Test accuracy (%) of models trained on data protected by different defensive noises via adversarial training with different perturbation radii. The defensive perturbation radius $\rho_u$ is set as $8/255$ for every type of noise, while the adversarial perturbation radius $\rho_a$ of REM noise takes various values.

| Dataset | Adv. Train. $\rho_a$ | Clean | EM | TAP | NTGA | REM $\rho_a = 0$ | 1/255 | 2/255 | 3/255 | 4/255 |
|---|---|---|---|---|---|---|---|---|---|---|
| CIFAR-10 | 0 | 94.66 | 13.20 | 22.51 | 16.27 | 15.18 | **13.05** | 20.60 | 20.67 | 27.09 |
| | 1/255 | 93.74 | 22.08 | 92.16 | 41.53 | 27.20 | **14.28** | 22.60 | 25.11 | 28.21 |
| | 2/255 | 92.37 | 71.43 | 90.53 | 85.13 | 75.42 | 29.78 | **25.41** | 27.29 | 30.69 |
| | 3/255 | 90.90 | 87.71 | 89.55 | 89.41 | 88.08 | 73.08 | 46.18 | **30.85** | 35.80 |
| | 4/255 | 89.51 | 88.62 | 88.02 | 88.96 | 89.15 | 86.34 | 75.14 | **47.51** | 48.16 |
| CIFAR-100 | 0 | 76.27 | **1.60** | 13.75 | 3.22 | 1.89 | 3.72 | 3.03 | 8.31 | 10.14 |
| | 1/255 | 71.90 | 71.47 | 70.03 | 65.74 | 9.45 | **4.47** | 5.68 | 9.86 | 11.99 |
| | 2/255 | 68.91 | 68.49 | 66.91 | 66.53 | 52.46 | 13.36 | **7.03** | 11.32 | 14.15 |
| | 3/255 | 66.45 | 65.66 | 64.30 | 64.80 | 66.27 | 44.93 | 27.29 | **17.55** | 17.74 |
| | 4/255 | 64.50 | 63.43 | 62.39 | 62.44 | 64.17 | 61.70 | 61.88 | 41.43 | **27.10** |
| ImageNet Subset | 0 | 80.66 | **1.26** | 9.10 | 8.42 | 2.54 | 4.52 | 6.20 | 8.88 | 13.74 |
| | 1/255 | 76.20 | 74.88 | 75.14 | 63.28 | 12.80 | 14.68 | **13.42** | 14.92 | 21.58 |
| | 2/255 | 72.52 | 71.74 | 70.56 | 66.96 | 49.58 | 33.14 | 27.06 | **23.76** | 29.40 |
| | 3/255 | 69.68 | 66.90 | 67.64 | 65.98 | 66.68 | 42.97 | 41.18 | **32.16** | 35.76 |
| | 4/255 | 66.62 | 63.40 | 63.56 | 63.06 | 64.80 | 59.32 | 51.78 | 41.52 | **41.66** |

**Model training.** We follow Eq. (2) to conduct adversarial training (Madry et al., 2018) on data protected by different defensive noises with different models, including VGG-16 (Simonyan & Zisserman, 2014), ResNet-18, ResNet-50 (He et al., 2016), DenseNet-121 (Huang et al., 2017), and wide ResNet-34-10 (Zagoruyko & Komodakis, 2016). Similar to that in training noise generator, we also focus on $L_\infty$-bounded noise $\|\rho_a\|_\infty \leq \rho_a$ in adversarial training. Note that when $\rho_a$ takes 0, the adversarial training in Eq. (2) degenerates to the ERM training in Eq. (1). The detailed training settings are given in Appendix A.4.

**Metric.** We use the test accuracy to assess the data protection ability of the defensive noise. A low test accuracy suggests that the model learned little knowledge from the training data, which thus implies a strong protection ability of the noise.

## 5.2 EFFECTIVENESS OF ROBUST ERROR-MINIMIZING NOISE

In this section, we study the effectiveness of robust error-minimizing noise from different aspects. We first visualize different defensive noises and the correspondingly crafted examples in Fig. 3. More visualization results can be found in Appendix B.

**Different adversarial training perturbation radius.** We first add different defensive noises to the entire training set. The defensive perturbation radius $\rho_u$ of every noise is set as $8/255$. Adversarial training is then conducted on both clean data and modified data with ResNet-18 models and different adversarial training perturbation radii $\rho_a$. Table 1 reports the accuracies of the trained models on clean test data. We have also conducted experiments on noises that generated with a larger defensive perturbation radius $\rho_u = 16/255$, and the results are presented in Table 7 in Appendix C.

Table 2: Test accuracy (%) on CIFAR-10 and CIFAR-100 with different protection percentages. For EM, TAP, and NTGA noises, the perturbation radius $\rho_u$ is set as $8/255$. For REM noise, the perturbation radii $\rho_u$ and $\rho_a$ are set as $8/255$ and $4/255$, respectively.

| Dataset | Adv. Train. $\rho_a$ | Noise Type | Data Protection Percentage | | | | | | | | | |
|---|---|---|---|---|---|---|---|---|---|---|---|---|
| | | | 0% | 20% | | 40% | | 60% | | 80% | | 100% |
| | | | | Mixed | Clean | Mixed | Clean | Mixed | Clean | Mixed | Clean | |
| CIFAR-10 | 2/255 | EM | 92.37 | 92.26 | 91.30 | 91.94 | 90.31 | 91.81 | 88.65 | 91.14 | 83.37 | 71.43 |
| | | TAP | | 92.17 | | 91.62 | | 91.32 | | 91.48 | | 90.53 |
| | | NTGA | | 92.41 | | 92.19 | | 92.23 | | 91.74 | | 85.13 |
| | | REM | | 92.36 | | 90.22 | | 88.45 | | 82.98 | | 30.69 |
| | 4/255 | EM | 89.51 | 89.60 | 88.17 | 89.40 | 86.76 | 89.49 | 85.07 | 89.10 | 79.41 | 88.62 |
| | | TAP | | 89.01 | | 88.66 | | 88.40 | | 88.04 | | 88.02 |
| | | NTGA | | 89.56 | | 89.35 | | 89.22 | | 89.17 | | 88.96 |
| | | REM | | 89.60 | | 89.34 | | 89.61 | | 88.09 | | 48.16 |
| CIFAR-100 | 2/255 | EM | 68.91 | 68.51 | 66.54 | 68.72 | 64.21 | 67.96 | 58.35 | 68.44 | 47.99 | 68.49 |
| | | TAP | | 68.05 | | 67.83 | | 67.75 | | 67.27 | | 66.91 |
| | | NTGA | | 68.52 | | 68.82 | | 68.36 | | 68.71 | | 66.53 |
| | | REM | | 69.00 | | 68.20 | | 60.75 | | 52.33 | | 14.15 |
| | 4/255 | EM | 64.50 | 63.86 | 61.73 | 64.24 | 57.61 | 63.62 | 53.86 | 63.37 | 44.79 | 63.43 |
| | | TAP | | 63.21 | | 63.01 | | 62.95 | | 62.90 | | 62.39 |
| | | NTGA | | 63.48 | | 63.59 | | 63.64 | | 62.83 | | 62.44 |
| | | REM | | 63.75 | | 63.99 | | 64.36 | | 63.05 | | 27.10 |

As shown in Table 1, when the adversarial training perturbation radius increases, the test accuracy of the model trained on unlearnable data and targeted adversarial poisoned data rapidly increases. On the other hand, the robust error-minimizing noise can always significantly reduces the test accuracy even when adversarial training presents. Furthermore, in the extreme case when the model trained on unlearnable data and targeted adversarial poisoned data achieve the same performance as that trained on clean data, the robust error-minimizing noise can still reduce the test accuracy of the model trained on robust unlearnable data by around $20\%$ to $40\%$. These results demonstrate the effectiveness of the robust error-minimizing noise against adversarial learners.

**Different protection percentages.** We then study a more challenging as well as more realistic learning scenario, where only a part of the data are protected by the defensive noise, while the others are clean. Specifically, we randomly select a part of the training data from the whole training set, adding defensive noise to the selected data, and conduct adversarial training with ResNet-18 on the mixed data and the remaining clean data. The defensive perturbation radius for every noise is set as $8/255$, while the adversarial perturbation radius $\rho_a$ of the robust error-minimizing noise is set as $4/255$. The accuracies on clean test data are reported in Table 2. The difference between the test accuracies on mixed data and clean data reflects the knowledge gained from the protected training data. We have also conducted experiments with noises that generated with a larger defensive perturbation radius $\rho_u = 16/255$. The results are reported in Table 8 in Appendix C.

Table 2 shows that when the adversarial training perturbation radius is small, robust error-minimizing noise can effectively protect the selected part of the data. However, when a large adversarial training perturbation radius presents, the protection becomes worthless. This suggests that to protect data against adversarial training with a perturbation radius $\rho_a$, one has to set the defensive perturbation radius $\rho_u$ of the robust error-minimizing noise to a value that is relatively larger than $\rho_a$. Table 2 shows that as the data protection percentage decreases, the performance of the trained model increases. This suggests that the model can learn more knowledge from more clean data, which coincides with intuition. Nevertheless, Table 2 further shows that the data protection ability of the robust error-minimizing noise is stronger than that of the error-minimizing noise and the targeted adversarial poisoning noise. This demonstrates that the robust error-minimizing noise is still more favorable than other types of defensive noise when adversarial training presents.

**Different model architectures.** So far, we have only conduct adversarial training with ResNet-18, which is as same as the source model in the defensive noise generation. We now evaluate the effectiveness of the robust error-minimizing noise under different adversarial learning models. Specifically, we conduct adversarial training with a perturbation radius $4/255$ and five different types of model, including VGG-16, ResNet-18, ResNet-50, DenseNet-121, and wide ResNet-34-10, on data that is protected by noise generated via ResNet-18. Table 3 presents the test accuracies of the

Table 3: Test accuracy (%) of different types of models on CIFAR-10 and CIFAR-100 datasets. The adversarial training perturbation radius is set as $4/255$. The defensive perturbation radius $\rho_u$ of every type of defensive noise is set as $8/255$.

| Dataset | Model | Clean | EM | TAP | NTGA | REM | |
|---------|-------|-------|-----|-----|------|-----|-----|
| | | | | | | $\rho_a = 2/255$ | $4/255$ |
| CIFAR-10 | VGG-16 | 87.51 | 86.48 | 86.27 | 86.65 | 75.97 | **65.23** |
| | RN-18 | 89.51 | 88.62 | 88.02 | 88.96 | 75.14 | **48.16** |
| | RN-50 | 89.79 | 89.28 | 88.45 | 88.79 | 73.59 | **40.65** |
| | DN-121 | 83.27 | 82.44 | 81.72 | 80.73 | **77.82** | 81.48 |
| | WRN-34-10 | 91.21 | 90.05 | 90.23 | 89.95 | 73.98 | **48.39** |
| CIFAR-100 | VGG-16 | 57.14 | 56.94 | 55.24 | 55.81 | 56.50 | **48.85** |
| | RN-18 | 63.43 | 64.17 | 62.39 | 62.44 | 61.88 | **27.10** |
| | RN-50 | 66.93 | 66.43 | 64.44 | 64.91 | 61.30 | **26.03** |
| | DN-121 | 53.73 | 53.52 | 52.93 | **52.40** | 54.19 | 54.48 |
| | WRN-34-10 | 68.64 | 68.27 | 65.80 | 67.41 | 64.11 | **25.04** |

trained models on CIFAR-10 and CIFAR-100. We have also conducted experiments on noises that generated with a larger defensive perturbation radius $\rho_u = 16/255$. See Table 9 in Appendix C.

Table 3 shows the robust error-minimizing noise generated from ResNet-18 can effectively protect data against various adversarially trained models. However, when the DenseNet-121 model presents, the robust error-minimizing noise could not achieve the same protection performance as that in other scenarios. This may partially be attributed to the limitation of DenseNet-121 itself in adversarial training, as it could not achieve the same generalization ability as those other models on clean data. We will leave further studies on this phenomenon in future works. Nevertheless, the experiment results still demonstrate the effectiveness of robust error-minimizing noise in most cases.

**Ablation study on the EOT technique.** In Section 4.3, we propose to employ the EOT technique to enhance the stability of the robust error-minimizing noise in adversarial training. Here, we empirically investigate the effect of EOT. Specifically, we conduct adversarial training with ResNet-18 on datasets protected by robust error noises that are generated with and without the EOT technique, respectively. The accuracies of the trained models on clean test data are reported in Table 4. The results show that without EOT, the gener-

Table 4: Ablation study on the EOT technique. Test accuracies (%) of models trained on CIFAR-10 and CIFAR-100 are reported. The defensive perturbation radius $\rho_u$ of the REM noise is set as $8/255$. "EOT" and "None" denote that the noise is generated with and without the EOT technique, respectively.

| Dataset | Adv. Train. $\rho_a$ | Clean | REM | | | |
|---------|-----------------------|-------|-----|------|-----|------|
| | | | $\rho_a = 2/255$ | | $4/255$ | |
| | | | EOT | None | EOT | None |
| CIFAR-10 | 0/255 | 94.66 | **20.60** | 29.15 | 27.09 | **18.55** |
| | 2/255 | 92.37 | **25.41** | 91.69 | **30.69** | 91.76 |
| | 4/255 | 89.51 | **75.14** | 89.17 | **48.16** | 89.41 |
| CIFAR-100 | 0/255 | 76.27 | **3.03** | 4.64 | 10.14 | **7.03** |
| | 2/255 | 68.91 | **7.03** | 68.85 | **14.15** | 69.98 |
| | 4/255 | 64.50 | **61.88** | 63.51 | **27.10** | 63.79 |

ated noise gains no protection ability, which in turn justifies the necessity of employing EOT during the robust error-minimizing noise generation.

# 6 CONCLUSION AND FUTURE WORKS

This paper proposes robust error minimizing noise, the first defensive noise that can protect data from being learned by unauthorized adversarial training. We deduce that the error-minimizing noise proposed by Huang et al. (2021) could not prevent unauthorized adversarial training mainly because it can not suppress the training loss in adversarial training based on empirical study. Motivated by this, the robust error-minimizing noise is introduced to effectively reduce the adversarial training loss and thus suppress the learnable knowledge of data in adversarial training. Inspired by the adversarial training loss and the vanilla error-minimizing noise generation, a min-min-max problem is designed for training the robust error-minimizing noise generator, where the expectation over transform technique is adopted to enhance the stability of the generated noise. An important future direction is to establish theoretical foundations for the effectiveness of robust-error minimizing noise. Another interesting direction is to design more efficient robust error-minimizing noise generation methods.

ACKNOWLEDGMENTS

This work is supported by the Major Science and Technology Innovation 2030 "New Generation Artificial Intelligence" key project (No. 2021ZD0111700). Yang Liu is supported in part by the Tsinghua (AIR)-Asiainfo Technologies (China) Research Center under grant No. 20203910074. The authors sincerely appreciate the anonymous ICLR reviewers for their helpful comments.

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

# A    EXPERIMENT DETAILS

This section provides the experiment details omitted from Section 5.

## A.1    HARDWARE DETAILS

The experiments on CIFAR-10 and CIFAR-100 are conducted on 1 GPU (NVIDIA® Tesla® V100 16GB) and 10 CPU cores (Intel® Xeon® Processor E5-2650 v4 @ 2.20GHz).

The experiments on ImageNet are conducted on 4 GPU (NVIDIA® Tesla® V100 16GB) and 40 CPU cores (Intel® Xeon® Processor E5-2650 v4 @ 2.20GHz).

## A.2    DATA AUGMENTATION

We use different data augmentations for different datasets. For CIFAR-10 and CIFAR-100, we perform data augmentation via random flipping, padding $4$ pixels on each side, random cropping to $32 \times 32$ size, and rescaling per pixel to $[-0.5, 0.5]$ for each image. For the ImageNet subset, we perform data augmentation via random cropping, resizing to $224 \times 224$ size, random flipping, and rescaling per pixel to $[-0.5, 0.5]$ for each image.

## A.3    DEFENSIVE NOISE GENERATION

Our experiments involve three types of defensive noise, the proposed robust error-minimizing noise, and two baseline methods, error-minimizing noise and adversarial poisoning noise.

Table 5: The settings of PGD (see Eq. (3)) for the noise generations of error-minimizing noise (EM), targeted adversarial poisoning noise (TAP), neural tangent generalization attack noise (NTGA), and robust error-minimizing noise (REM) in different experiments. $\rho_u$ denotes the defensive perturbation radius of different types of noise, while $\rho_a$ denotes the adversarial perturbation radius of the robust error-minimizing noise.

| Datasets | Noise Type | $\alpha_u$ | $K_u$ | $\alpha_a$ | $K_a$ |
|---|---|---|---|---|---|
| CIFAR-10 CIFAR-100 | EM | $\rho_u/5$ | 10 | - | - |
| | TAP | $\rho_u/125$ | 250 | - | - |
| | NTGA | $\rho_u/10 \times 1.1$ | 10 | - | - |
| | REM | $\rho_u/5$ | 10 | $\rho_a/5$ | 10 |
| ImageNet Subset | EM | $\rho_u/5$ | 7 | - | - |
| | TAP | $\rho_u/50$ | 100 | - | - |
| | NTGA | $\rho_u/8 \times 1.1$ | 8 | - | - |
| | REM | $\rho_u/4$ | 7 | $\rho_a/5$ | 10 |

### A.3.1    ROBUST ERROR-MINIMIZING NOISE

Following Huang et al. (2021), we employ ResNet-18 (He et al., 2016) as the source model $f'$ for the robust error-minimizing noise generation. The $L_\infty$-bounded noises $\|\delta_u\|_\infty \leq \rho_u$ and $\|\delta_a\|_\infty \leq \rho_a$ are adopted in our experiments, in which the defensive perturbation radius $\rho_u$ and adversarial perturbation radius $\rho_a$ can take various values. The settings of PGD for solving the inner minimization and maximization problems in Eq. (5) are presented in Table 5.

For CIFAR-10 and CIFAR-100, each source model is trained with SGD for $5,000$ iterations, with a batch size of $128$, a momentum factor of $0.9$, a weight decay factor of $0.0005$, an initial learning rate of $0.1$, and a learning rate scheduler that decay the learning rate by a factor of $0.1$ every $2,000$ iterations. For EOT, the data transformation $T$ is set as the data augmentation of the corresponding dataset, and the repeatedly sampling number for expectation estimation is set as $5$.

Besides, for the ImageNet subset, each source model is trained with SGD via Eq. (5) for $3,000$ iterations, with a batch size of $128$, a momentum factor of $0.9$, a weight decay factor of $0.0005$, an initial learning rate of $0.1$, and a learning rate scheduler that decay the learning rate by a factor of $0.1$ every $1,200$ iterations. For EOT, the data transformation $T$ is set as the data augmentation of the corresponding dataset, and the repeatedly sampling number for expectation estimation is set as $4$.

### A.3.2 BASELINE METHODS IMPLEMENTATIONS

Two baseline methods are adopted in our experiments as comparisons, including the error-minimizing noise method and the targeted adversarial poisoning noise method. Every method is reproduced on our own.

**Error-minimizing noise (Huang et al., 2021).** We follow Eq. (3) to train the error-minimizing noise generator. The error-minimizing noise is then generated with the trained noise generator. ResNet-18 model is used as the source model $f'$. PGD is employed for solving the inner minimization problem in Eq. (3), where the settings of PGD is presented in Table 5. Other hyperparameters for training the noise generator are set the same as that for the robust error-minimizing noise generator in the previous section.

**Targeted adversarial poisoning noise (Fowl et al., 2021b).** This type of noise is generated via conducting targeted adversarial attack to the model that is trained on clean data, in which the generated adversarial perturbation is used as the adversarial poisoning noise. Specifically, given a fixed model $f_0$ and a sample $(x, y)$, the targeted adversarial attack will generate noise via solving the problem $\arg\max_{\|\delta_u\| \leq \rho_u} \ell(f_0(x + \delta_u), g(y))$, where $g$ is a permutation function on the label space $\mathcal{Y}$. PGD with *differentiable data augmentation* (Geiping et al., 2021) is employed for solving the above problem. The hyper-parameters for the PGD is given in Table 5.

**Neural tangent generalization attack noise (Yuan & Wu, 2021).** This type of protective noise aims to weaken the generalization ability of the model trained on the modified data. To do this, an ensemble of neural networks is modeled based on neural tangent kernel (NTK) (Jacot et al., 2018), and the NTGA noise is generated upon the ensemble model. As a result, the NTGA noise enjoys remarkable transferability. We use the official code of NTGA [1] to generate this type of noise. Specifically, we employ the FNN model in Yuan & Wu (2021) as the ensemble model. For CIFAR-10 and CIFAR-100, the block size for approximating NTK is set as $4,000$. For the ImageNet subset, the block size is set as $100$. The hyper-parameters for the PGD are given in Table 5. Other experiment settings follow Yuan & Wu (2021), please refer accordingly.

### A.3.3 TIME COSTS OF NOISE GENERATIONS

We calculate the time costs for training different types of noise generators on different datasets. The results are reported in the following Table 6.

Table 6: Time costs for training different defensive noise generators on different datasets.

| Dataset | EM | TAP | NTGA | REM |
|---|---|---|---|---|
| CIFAR-10 | 0.4h | 0.5h | 5.2h | 22.6h |
| CIFAR-100 | 0.4h | 0.5h | 5.2h | 22.6h |
| ImageNet Subset | 3.9h | 5.2h | 14.6h | 51.2h |

### A.4 MODEL TRAINING DETAILS

We follow Eq. (2) to perform adversarial training (Madry et al., 2018). Similar to that in training noise generator, we also focus on $L_\infty$-bounded noise $\|\rho_a\|_\infty \leq \rho_a$ in adversarial training.

In every experiment, the model is trained with SGD for $40,000$ iterations, with a batch size of $128$, a momentum factor of $0.9$, a weight decay factor of $0.0005$, an initial learning rate of $0.1$, and a learning rate scheduler that decays the learning rate by a factor of $0.1$ every $16,000$ iterations. For CIFAR-10 and CIFAR-100, the steps number $K_a$ and the step size $\alpha_a$ in PGD are set as $10$ and $\rho_a/5$. For the ImageNet subset, the steps number $K_a$ and the step size $\alpha_a$ are set as $8$ and $\rho_a/4$.

---

[1]The official code of NTGA is available at `https://github.com/lionelmessi6410/ntga`.

# B    NOISE VISUALIZATION

This section presents more visualization results of different defensive noises.

## B.1    CIFAR-10

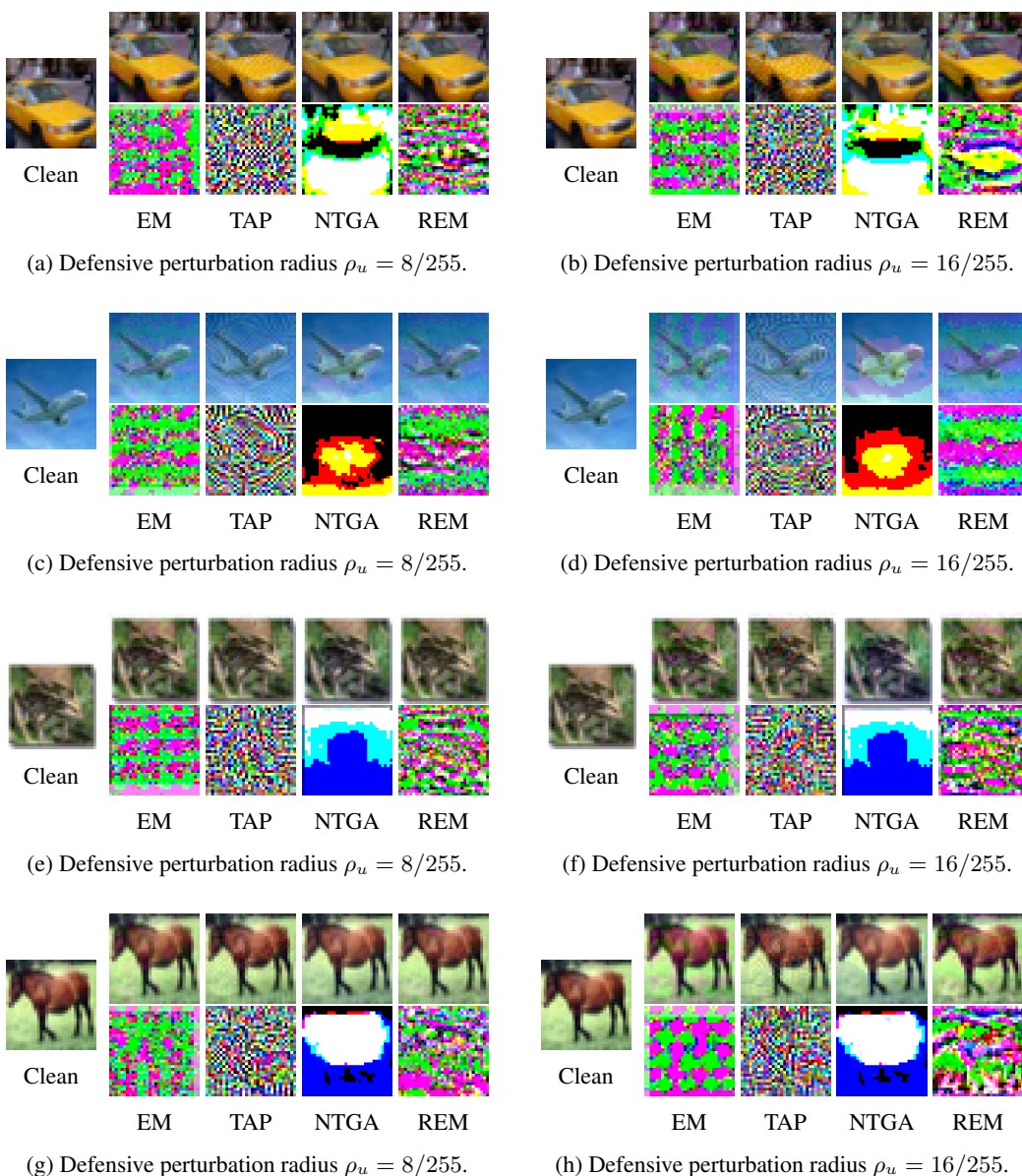

(a) Defensive perturbation radius $\rho_u = 8/255$.    (b) Defensive perturbation radius $\rho_u = 16/255$.

(c) Defensive perturbation radius $\rho_u = 8/255$.    (d) Defensive perturbation radius $\rho_u = 16/255$.

(e) Defensive perturbation radius $\rho_u = 8/255$.    (f) Defensive perturbation radius $\rho_u = 16/255$.

(g) Defensive perturbation radius $\rho_u = 8/255$.    (h) Defensive perturbation radius $\rho_u = 16/255$.

Figure 4: Visualization results of CIFAR-10. Examples of data protected by error-minimizing noise (EM), targeted adversarial poisoning noise (TAP), neural tangent generalization attack noise (NTGA), and robust error-minimizing noise (REM). When $\rho_u$ is set as $8/255$ or $16/255$, the adversarial perturbation radius $\rho_a$ of REM is set as $4/255$ or $8/255$.

## B.2 CIFAR-100

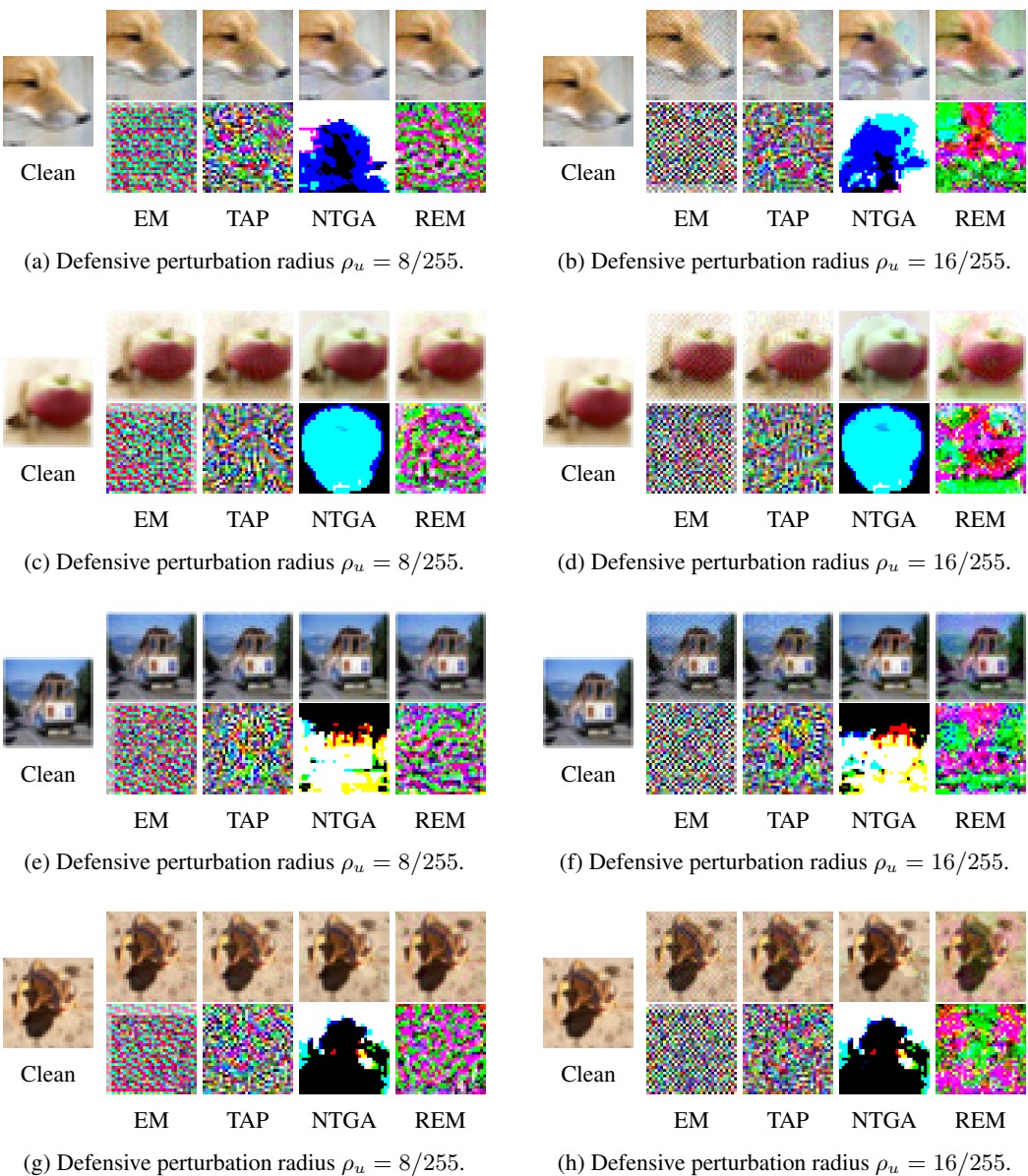

Figure 5: Visualization results of CIFAR-100. Examples of data protected by error-minimizing noise (EM), targeted adversarial poisoning noise (TAP), neural tangent generalization attack noise (NTGA), and robust error-minimizing noise (REM). When $\rho_u$ is set as $8/255$ or $16/255$, the adversarial perturbation radius $\rho_a$ of REM is set as $4/255$ or $8/255$.

## B.3 IMAGENET SUBSET

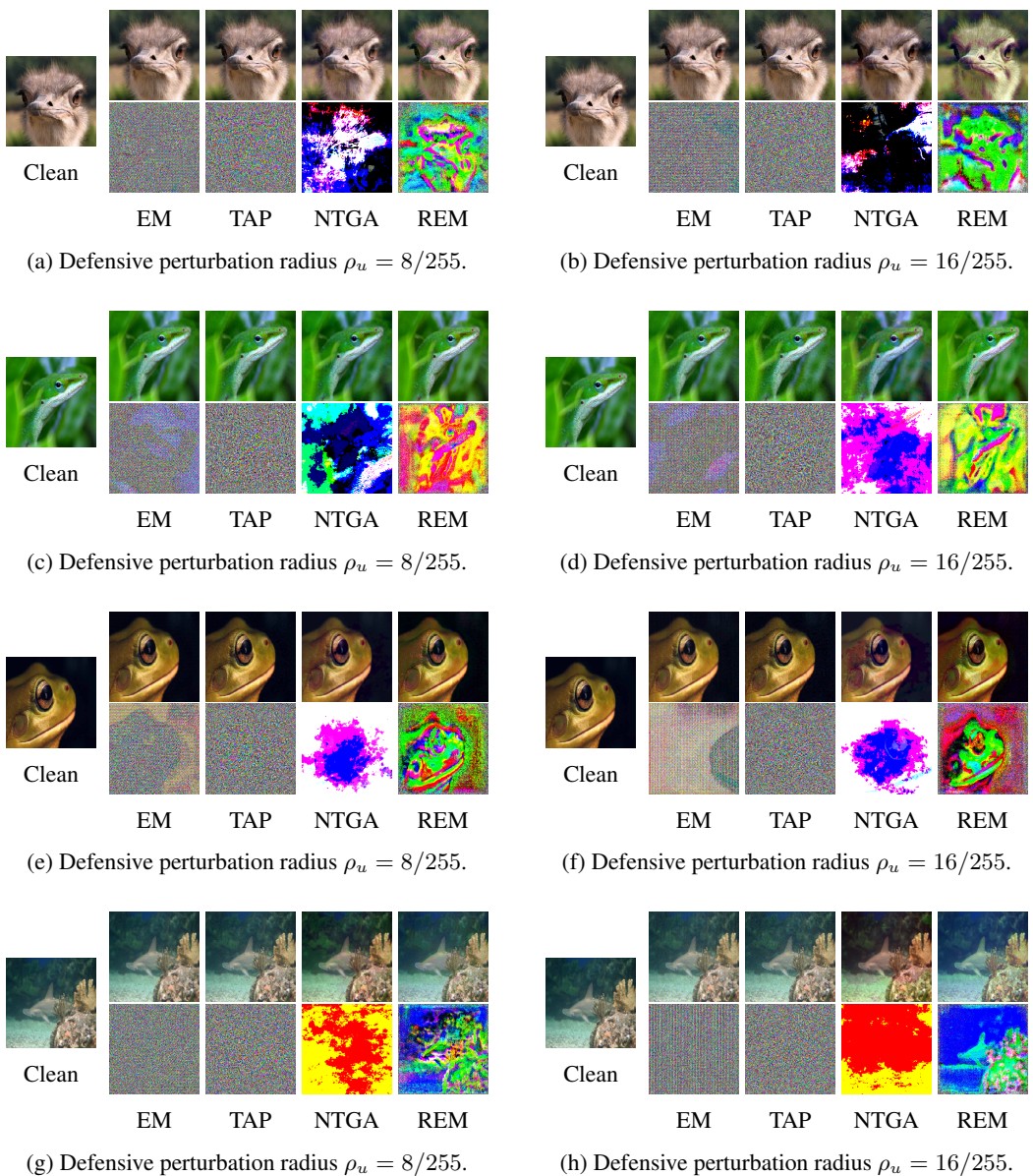

Figure 6: Visualization results of ImageNet subset. Examples of data protected by error-minimizing noise (EM), targeted adversarial poisoning noise (TAP), neural tangent generalization attack noise (NTGA), and robust error-minimizing noise (REM). When $\rho_u$ is set as $8/255$ or $16/255$, the adversarial perturbation radius $\rho_a$ of REM is set as $4/255$ or $8/255$.

## C  MORE EXPERIMENTS RESULTS

This section collects the additional experiment results for Section 5.2.

**Different adversarial training perturbation radius.**

Table 7: Test accuracy (%) of models trained on data that are protected by different defensive noises via adversarial training with different perturbation radii. The defensive perturbation radius $\rho_u$ is set as $16/255$ for every type of noise, while the adversarial perturbation radius $\rho_a$ of REM noise takes various values.

| Dataset | Adv. Train. $\rho_a$ | Clean | EM | TAP | NTGA | REM $\rho_a = 0$ | 2/255 | 4/255 | 6/255 | 8/255 |
|---|---|---|---|---|---|---|---|---|---|---|
| | 0 | 94.66 | 16.84 | 11.29 | **10.91** | 13.69 | 13.15 | 19.51 | 24.54 | 24.08 |
| | 2/255 | 92.37 | 22.46 | 78.01 | 19.96 | 21.17 | **17.73** | 20.46 | 21.89 | 26.30 |
| CIFAR-10 | 4/255 | 89.51 | 41.95 | 87.60 | 32.80 | 45.87 | 31.18 | **23.52** | 26.11 | 28.31 |
| | 6/255 | 86.90 | 52.13 | 85.44 | 60.64 | 66.82 | 58.37 | 40.25 | **30.93** | 31.50 |
| | 8/255 | 84.79 | 64.71 | 82.56 | 74.50 | 77.08 | 72.87 | 63.49 | 46.92 | **36.37** |
| | 0 | 76.27 | **1.44** | 4.84 | 1.54 | 2.14 | 4.16 | 4.27 | 5.86 | 11.16 |
| | 2/255 | 68.91 | 5.21 | 64.59 | 5.21 | 6.68 | 5.04 | **4.83** | 7.86 | 13.42 |
| CIFAR-100 | 4/255 | 64.50 | 35.65 | 61.48 | 18.43 | 28.27 | 9.80 | **6.87** | 9.06 | 14.46 |
| | 6/255 | 60.86 | 56.73 | 57.66 | 46.30 | 47.08 | 35.25 | 30.16 | **14.41** | 17.67 |
| | 8/255 | 58.27 | 56.66 | 55.30 | 50.81 | 54.23 | 49.82 | 54.55 | 33.86 | **23.29** |
| | 0 | 80.66 | **1.10** | 3.96 | 3.42 | 2.90 | 3.40 | 3.86 | 5.90 | 9.72 |
| | 2/255 | 72.52 | 70.94 | 70.80 | 19.90 | 16.00 | 8.50 | **5.66** | 8.78 | 14.74 |
| ImageNet Subset | 4/255 | 66.62 | 62.16 | 62.54 | 41.08 | 44.12 | 20.58 | **11.20** | 12.46 | 19.38 |
| | 6/255 | 58.80 | 55.46 | 54.68 | 44.32 | 55.04 | 34.04 | 19.12 | **16.24** | 23.14 |
| | 8/255 | 53.12 | 46.44 | 46.90 | 42.78 | 47.96 | 44.54 | 28.20 | **21.66** | 26.64 |

**Different data protection percentages.**

Table 8: Test accuracy (%) of models trained on CIFAR-10 and CIFAR-100 with different protection percentages. The defensive perturbation radius of every noise is set as $\rho_u = 16/255$. The adversarial perturbation radius of robust error-minimizing noise is set as $8/255$.

| Dataset | Adv. Train. $\rho_a$ | Noise Type | 0% | 20% mixed | 20% clean | 40% mixed | 40% clean | 60% mixed | 60% clean | 80% mixed | 80% clean | 100% |
|---|---|---|---|---|---|---|---|---|---|---|---|---|
| CIFAR-10 | 4/255 | EM | 89.51 | 89.45 | 88.17 | 88.00 | 86.76 | 86.53 | 85.07 | 81.62 | 79.41 | 41.95 |
| | | TAP | | 88.80 | | 88.64 | | 87.50 | | 87.83 | | 87.60 |
| | | NTGA | | 89.63 | | 88.70 | | 87.11 | | 83.32 | | 32.80 |
| | | REM | | 89.08 | | 87.00 | | 85.05 | | 79.00 | | 28.31 |
| | 8/255 | EM | 84.79 | 84.20 | 83.25 | 84.37 | 81.54 | 83.70 | 79.29 | 82.91 | 73.21 | 64.71 |
| | | TAP | | 84.24 | | 83.58 | | 83.42 | | 83.09 | | 82.56 |
| | | NTGA | | 84.62 | | 84.18 | | 84.48 | | 83.26 | | 74.50 |
| | | REM | | 84.85 | | 83.99 | | 83.15 | | 80.05 | | 36.37 |
| CIFAR-100 | 4/255 | EM | 64.50 | 64.27 | 61.73 | 63.26 | 57.61 | 63.86 | 53.86 | 61.45 | 44.79 | 35.65 |
| | | TAP | | 63.00 | | 62.69 | | 62.65 | | 60.99 | | 61.48 |
| | | NTGA | | 63.47 | | 63.09 | | 63.14 | | 58.72 | | 18.43 |
| | | REM | | 63.01 | | 59.90 | | 54.52 | | 47.27 | | 14.46 |
| | 8/255 | EM | 58.27 | 57.44 | 55.00 | 57.70 | 50.72 | 57.14 | 46.29 | 57.27 | 39.65 | 56.66 |
| | | TAP | | 57.75 | | 57.15 | | 56.32 | | 55.73 | | 55.30 |
| | | NTGA | | 57.49 | | 56.35 | | 55.69 | | 53.92 | | 50.81 |
| | | REM | | 57.78 | | 58.12 | | 57.69 | | 55.30 | | 23.29 |

**Different model architectures.**

Table 9: Test accuracy (%) of different types of models trained on CIFAR-10 and CIFAR-100. The defensive perturbation radius $\rho_u$ of every defensive noise is set as $16/255$. The adversarial training perturbation radius is set as $8/255$.

| Dataset | Model | Clean | EM | TAP | NTGA | REM $\rho_a = 4/255$ | REM $8/255$ |
|---------|-------|-------|-----|-----|------|------|------|
| CIFAR-10 | VGG-16 | 79.92 | 63.18 | 77.73 | 69.99 | 63.02 | **41.17** |
| | RN-18 | 84.79 | 64.71 | 82.56 | 74.50 | 63.49 | **36.37** |
| | RN-50 | 79.92 | 61.32 | 82.59 | 72.39 | 57.84 | **31.91** |
| | DN-121 | 75.75 | 67.70 | 74.35 | 66.56 | 62.39 | **58.70** |
| | WRN-34-10 | 86.36 | 65.42 | 84.19 | 76.42 | 64.91 | **37.51** |
| CIFAR-100 | VGG-16 | 45.67 | 45.56 | 43.42 | 38.83 | 48.44 | **22.54** |
| | RN-18 | 58.27 | 56.66 | 55.30 | 50.81 | 54.55 | **23.29** |
| | RN-50 | 60.15 | 58.34 | 56.67 | 51.88 | 54.68 | **21.19** |
| | DN-121 | 48.28 | 47.12 | 46.74 | **41.05** | 47.36 | 42.87 |
| | WRN-34-10 | 60.95 | 58.50 | 57.92 | 54.47 | 58.22 | **22.67** |

**The evolution of the train and test accuracies along the adversarial training process.** The curves of train and test accuracies on data protected by different defensive noises are drawn in Fig. 7. The figure shows that robust error-minimizing noise can effectively protect data from being learned along the adversarial training process, while other existing approaches could not. This result further justifies the effectiveness of our proposed method.

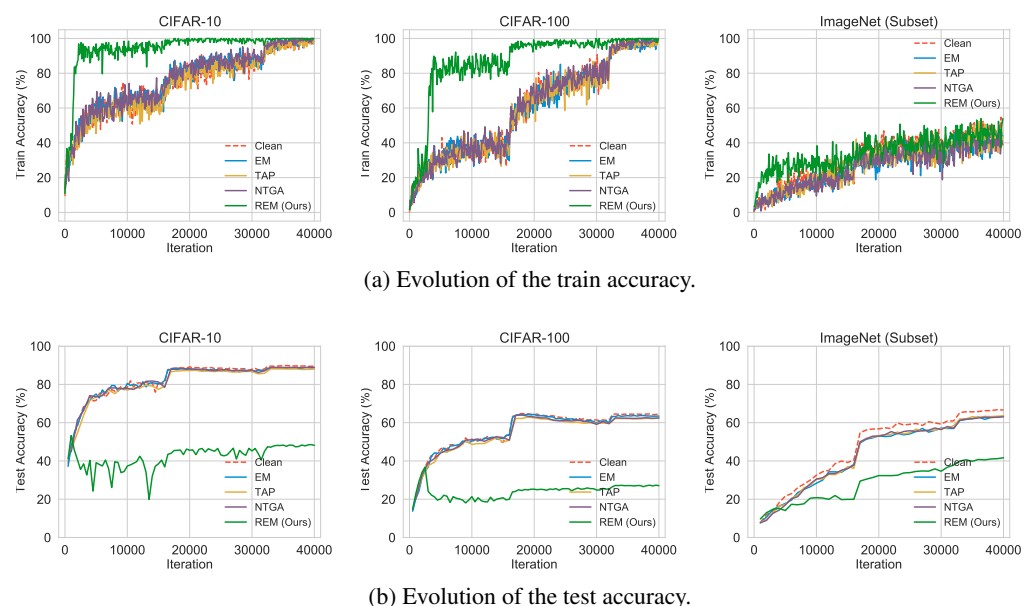

(a) Evolution of the train accuracy.

(b) Evolution of the test accuracy.

Figure 7: The curves of train and test accuracies to training iteration on data protected by different defensive noises. The defensive perturbation radius $\rho_u$ for every noise is set as $8/255$, while the adversarial perturbation radius $\rho_a$ for REM is set as $4/255$. Besides, the adversarial training perturbation radius is set as $4/255$ in every experiment.

**The selections of adversarial perturbation radius $\rho_a$ in robust error-minimizing noise generation.** We conduct adversarial training against the robust error-minimizing noises generated with a fixed defensive perturbation radius $\rho_u$ and different adversarial perturbation radii $\rho_a$. The results are presented Fig. 8, which suggests that setting the adversarial perturbation radius to be half of the defensive perturbation radius would help the noise achieve consistent protection ability against adversarial training with different perturbation radii.

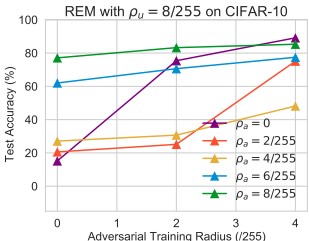 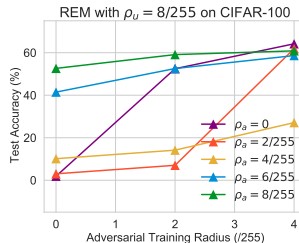

Figure 8: We conduct adversarial training on data protected by different robust error-minimizing noise (REM). The curves of test accuracy vs. adversarial training radius are plotted. The defensive perturbation radius $\rho_u$ for every REM noise is set as $8/255$.

## D  RESISTANCE TO LOW-PASS FILTERING

This section analyzes the resistance of different protective noises against low-pass filters. We first process each image from the training dataset (can be a clean dataset or protected dataset) with three low-pass filters, mean filter, median filter, and Gaussian filter, respectively. For every filter, the shape of the window is set as $3 \times 3$. We then conduct adversarial training on the processed data with various adversarial training perturbation radii. The experiment results are presented in Table 10.

From the table, we find that when the adversarial training perturbation is small, robust error-minimizing noise can effectively prevent data from being learned. However, when large adversarial training perturbation presents, even robust error-minimizing noise becomes ineffective. Nevertheless, compared with other types of noise, the protection brought by robust error-minimizing noise is stronger in most situations. These results demonstrate that robust error-minimizing is more favorable in protecting data against adversarial learning.

Table 10: Test accuracy (%) of different types of models trained on CIFAR-10 and CIFAR-100 that are processed by different low-pass filters. The defensive perturbation radius $\rho_u$ of every defensive noise is set as $16/255$. The adversarial training perturbation radius is set as $8/255$.

| Dataset | Filter | Adv. Train. $\rho_a$ | Clean | EM | TAP | NTGA | REM |
|---|---|---|---|---|---|---|---|
| CIFAR-10 | Mean | 2/255 | 84.25 | 34.87 | 82.53 | 40.26 | **28.60** |
| | | 4/255 | 80.47 | 56.41 | 78.87 | 57.42 | **40.32** |
| | | 8/255 | 74.74 | 73.42 | 72.98 | **67.69** | 68.70 |
| | Median | 2/255 | 87.04 | 31.86 | 85.10 | 30.87 | **27.36** |
| | | 4/255 | 83.87 | 49.33 | 82.31 | 48.50 | **34.06** |
| | | 8/255 | 78.34 | 74.02 | 76.66 | 67.63 | **62.14** |
| | Gaussian | 2/255 | 86.78 | 29.71 | 85.44 | 41.85 | **28.70** |
| | | 4/255 | 83.33 | 52.47 | 81.83 | 58.14 | **35.13** |
| | | 8/255 | 77.30 | 73.84 | 75.98 | 70.06 | **68.19** |
| CIFAR-100 | Mean | 2/255 | 52.42 | 53.07 | 51.30 | 26.49 | **13.89** |
| | | 4/255 | 50.89 | 50.61 | 50.35 | 35.46 | **24.52** |
| | | 8/255 | 47.73 | 46.98 | 46.62 | **42.03** | 44.96 |
| | Median | 2/255 | 57.69 | 56.35 | 55.22 | 18.14 | **14.08** |
| | | 4/255 | 54.77 | 53.50 | 53.31 | 33.05 | **19.14** |
| | | 8/255 | 51.09 | 49.64 | 48.86 | 45.13 | **40.45** |
| | Gaussian | 2/255 | 56.64 | 56.49 | 55.19 | 29.05 | **13.74** |
| | | 4/255 | 53.44 | 53.62 | 53.17 | 37.39 | **22.92** |
| | | 8/255 | 49.75 | 49.61 | 48.59 | **44.55** | 47.03 |

# E    RESISTANCE TO ADVERSARIAL TRAINING WITH DIFFERENT PERTURBATION NORMS

This section studies the resistance of different protective noises against adversarial training with various perturbation norms. Specifically, when perform adversarial training (*i.e.*, Eq. (2)) on the protected dataset, we set the perturbation norm Eq. (2) as $L_1$-norm or $L_2$-norm. Other training settings are as same as that presented in Appendix A.4. The experiment results are collected and presented in Table 11, which shows that robust error-minimizing noise can effectively protect data from adversarial training in these situations.

Table 11: Test accuracy (%) of different types of models trained on CIFAR-10 and CIFAR-100. The adversarial training is conducted with different types of perturbation norms. The defensive perturbation radius $\rho_u$ of every defensive noise is set as $16/255$. The adversarial training perturbation radius is set as $8/255$.

| Dataset | Norm Type | Adv. Train. $\rho_a$ | Clean | EM | TAP | NTGA | REM |
|---------|-----------|----------------------|-------|------|------|------|------|
| CIFAR-10 | $L_1$-norm | 1000/255 | 93.61 | **18.49** | 53.74 | 46.78 | 29.07 |
| | | 3000/255 | 90.40 | 69.82 | 89.65 | 84.69 | **55.98** |
| | $L_2$-norm | 50/255 | 92.66 | 44.11 | 91.30 | 73.00 | **32.90** |
| | | 100/255 | 89.91 | 82.55 | 88.70 | 86.75 | **62.76** |
| CIFAR-100 | $L_1$-norm | 1000/255 | 72.32 | 72.03 | 65.13 | 47.55 | **14.49** |
| | | 3000/255 | 67.37 | 66.77 | 65.64 | 64.83 | **40.84** |
| | $L_2$-norm | 50/255 | 70.05 | 69.52 | 67.45 | 65.31 | **17.43** |
| | | 100/255 | 65.65 | 65.25 | 63.86 | 63.31 | **47.16** |

