# OpenReview forum: "Robust Unlearnable Examples: Protecting Data Privacy Against Adversarial Learning"
_ICLR.cc/2022/Conference — ICLR 2022 Poster_

### Official Review · Reviewer_z2Tr · 2021-10-30

**Correctness:** 4
**Technical Novelty And Significance:** 3
**Empirical Novelty And Significance:** 4
**Recommendation:** 8
**Confidence:** 5

**Main Review:**

Overall, I lean towards recommending accepting this paper.

---

Strengths:
- The proposed robust unlearnable example is technically sound and empirically demonstrated it can be effective when used against adversarial training. Experiments are comprehensive.
- As suggested in the paper, none of the existing work can effectively be used to protect the data against adversarial training, this work demonstrated adversarial training could also be vulnerable to training-time perturbation/data poisoning.
- Paper's presentation is easy to follow and most related works have been discussed.

---

Question/weakness:
- How important is the EOT for the robust error-minimizing noise? An ablation study would be very helpful. Why is the REM on ImageNet can show the content of the image while not on CIFAR-10? Is this because of different data augmentations between CIFAR/ImageNet for EOT?
- Is the proposed method only effective if the epsilon for the defensive perturbation is larger than epsilon for the adversarial training? For table 1, if adversarial training uses larger epsilon, at what point does the REM become ineffective?
- What is extra overhead (time) compared to EM/TAP?
- Need some clarifications for Algorithm 1. Line 10-13 finds the adversarial perturbation for J steps, and line 14-15 finds the unlearnable perturbation for a single step, is that right? For the adversarial perturbation, then the maximum change is 4/255, but for a single step of the unlearnable perturbation, the step size is $\alpha_u$, wouldn't the adversarial perturbation always overwrite the unlearnable perturbation?

**Summary Of The Paper:**

This paper studies how to make data unlearnable towards adversarial training. In prior works, including dataset poisoning, no methods can prevent data from being learned in adversarial training. This paper proposed a robust error-minimizing noise to solve this task.

**Summary Of The Review:**

Existing works have demonstrated adversarial training (AT) can remove the effect of both data poisoning using targeted adversarial attacks and unlearnable examples. This paper demonstrated an effective method that can prevent the data from being used for AT. Despite the potential limitation on the perturbation radius difference between AT and the unlearnable, this work reveals a weakness of adversarial training.

---

> ### Author Response · Authors · 2021-11-14
> **To Reviewer z2Tr**
>
> Thank you for your thorough review and constructive comments. All your concerns have been carefully responded below. The manuscript is carefully revised accordingly. We sincerely hope our responses fully address your questions.
>
>
> **Q1.1:** _How important is the EOT for the robust error-minimizing noise? An ablation study would be very helpful._
>
> **A1.1:** Thanks for the suggestion. We have added an ablation study on EOT in the revised submission (in Section 5.2). The results are shown in the following table:
>
> | Dataset | REM with EOT | REM without EOT |
> | ---- | ---- | ---- |
> | CIFAR-10   | 48.16 | 89.41 |
> | CIFAR-100 | 27.10 | 63.79 |
>
> ***Table:** Test accuracy (%) of models trained on data protected by REM that is generated with or without the EOT technique. The adversarial training perturbation radius is set as $4/255$. The defensive perturbation radius $\rho_u$ and the adversarial perturbation radius $\rho_a$ for REM are set as $8/255$ and $4/255$, respectively.*
>
> This table suggests that EOT significantly increases the performance of REM.
>
>
>
> **Q1.2:** _Why is the REM on ImageNet can show the content of the image while not on CIFAR-10? Is this because of different data augmentations between CIFAR/ImageNet for EOT?_
>
> **A1.2:** It is a very interesting question! Our insight is the examples in ImageNet are high-resolution, thus REM can learn patterns that aligns well with the robust shapes.
>
>
> **Q2.1:** _Is the proposed method only effective if the epsilon for the defensive perturbation is larger than epsilon for the adversarial training? For table 1, if adversarial training uses larger epsilon, at what point does the REM become ineffective?_
>
> **A2.1:** Yes. We empirically find that defensive epsilon in REM needs to be two times larger than the perturbation radius in adversarial training. Thus, we need to set a sufficiently large defense budget in practice to protect the data from unauthorized exploitation. As for the case in Table 1, REM can protect adversarial training with perturbation radius no larger than $4/255$.
>
>
> **Q3:** _What is extra overhead (time) compared to EM/TAP?_
>
> **A3:** The overhead time compared to EM/TAP/REM is as follows,
>
> | Dataset | EM | TAP | REM |
> | ---- | ---- | ---- | ---- |
> | CIFAR-10 | 0.4 h | 0.5 h | 22.6 h |
> | CIFAR-100 | 0.4 h | 0.5 h | 22.6 h |
> | ImageNet (subset) | 3.9 h | 5.2 h | 51.2 h |
>
>
> **Q4.1:** _Need some clarifications for Algorithm 1. Line 10-13 finds the adversarial perturbation for J steps, and line 14-15 finds the unlearnable perturbation for a single step, is that right?_
>
> **A4.1:** Yes, it is right.
>
>
> **Q4.2:** _For the adversarial perturbation, then the maximum change is 4/255, but for a single step of the unlearnable perturbation, the step size is $\alpha_\mu$, wouldn't the adversarial perturbation always overwrite the unlearnable perturbation?_
>
> **A4.2:** We respectfully note that the adversarial perturbation would not overwrite the unlearnable perturbation. However, we remind that the adversarial perturbations found in Line 10-13 only serve to help find the proper gradient direction to optimize the unlearnable perturbation for one step ($K_u$ steps at all) in Line 14-15. Therefore, the adversarial perturbations would not overwrite the unlearnable perturbation.

---

> > ### Comment · Reviewer_z2Tr · 2021-11-25
> > **Thanks for addressing concerns.**
> >
> > After reading the response, my previous concerns/questions are addressed. Thus, I have increased my score to 8.

---

> > > ### Author Response · Authors · 2021-11-25
> > > **Thanks!**
> > >
> > > Thank you very much for recognizing our contributions and kind support!

---

### Official Review · Reviewer_XwVk · 2021-10-31

**Correctness:** 4
**Technical Novelty And Significance:** 2
**Empirical Novelty And Significance:** 3
**Recommendation:** 6
**Confidence:** 3

**Main Review:**

Strength
- The paper is well motivated as being able to securely release the dataset to the public is an important problem that people care about
- The authors proposed a sensible remedy (robust error minimization noise) for the weakness identified by the prior authors.
- The authors are thorough with the experiments to illustrate the effectiveness of the proposed approach compared to the prior methods
- The authors have provided sufficient details for the experiments to make me feel comfortable reproducing their results

Weakness
- The proposed method seems quite straightforward. While the method is effective, the results are not that surprising.

Questions
- In table 1, what are the differences between REM with radius 0 and EM? It seems like the numbers between the two columns are quite different



**Summary Of The Paper:**

Error minimization noise was first proposed so that when a dataset is released to the public, conventional empirical risk minimization cannot learn a good model from the perturbed dataset. However, error minimizing noise is only effective to the extent that adversarial training is not used. Adversarial training was identified as an effective method of overcoming the error minimization noise in prior work. In this paper, the authors proposed a variant of the error minimization noise, which continues to be robust even in the presence of adversarial training. Specifically, they generat the adversarial noise with a similar method as the original error minimization paper, but calculate the gradients with respect to the attacked input as opposed to the clean input.

**Summary Of The Review:**

While the novelty of the method is weaker, I do find the study of error minimizing noise to be important. The proposed method is sensible, and the experiments are quite thorough to convince me of the effectiveness of the proposed method. Overall, I like the submission, and would recommend weak accept.

---

> ### Author Response · Authors · 2021-11-14
> **To Reviewer XwVk**
>
> Thank you for your constructive comments and kind support. All your concerns have been carefully responded below. We sincerely hope our responses fully address your question.
>
>
> **Q1:** _The proposed method seems quite straightforward. While the method is effective, the results are not that surprising._
>
> **A1:** We agree that our method is natural and effective. It solves an important problem in a simple way. To our best knowledge, this is the first algorithm that can protect data from unauthorized exploitation from adversarial training.
>
>
> **Q2:** _In table 1, what are the differences between REM with radius 0 and EM? It seems like the numbers between the two columns are quite different._
>
> **A2:** Thanks. REM with radius 0 is considerably different from EM. When the radius is 0, REM degenerates to EM method with the expectation over transformation technique (EOT) which significantly improves the stability of the generated noise.
>
> Specifically, REM with radius 0 solves the following optimization problem,
> $$
> \min_{\theta} \frac{1}{n} \sum_{i=1}^n \min_{||\delta^u_i|| \leq \rho_u} \mathbb{E}_{t \sim T} \ell(f'_\theta(t(x_i)+\delta^u_i),y_i),
> $$
>
> while EM solves the following optimization problem,
> $$
> \min_{\theta} \frac{1}{n} \sum_{i=1}^n \min_{||\delta^u_i||\leq \rho_u}  \ell(f'_\theta(x_i+\delta^u_i),y_i).
> $$

---

### Official Review · Reviewer_ntwj · 2021-11-02

**Correctness:** 2
**Technical Novelty And Significance:** 2
**Empirical Novelty And Significance:** 2
**Recommendation:** 3
**Confidence:** 4

**Details Of Ethics Concerns:**

The paper studies how to prevent misuse of people's data. There is no ethics concern.

**Main Review:**

## Strengths
1. The paper is clearly written and is easy to follow.

2. The paper considers the adversarial learning setting, which might be useful in practice. The proposed min-min-max framework is reasonable in principle.

3. Extensive experiments are shown to demonstrate the effectiveness of the generated robust error-minimization noise in terms of reducing test performance (but it doesn't show how data privacy can be protected as discussed in the weakness).

## Weaknesses
1. The paper majorly builds upon the previous work on unlearnable examples which only consider normal training. Although this work improves the reliability of the noise generation by replacing the original loss with adversarial loss and considering the expectation of data augmentation, the proposed method is quite straightforward and a bit trivial. Therefore, the novelty of the paper is a bit limited.

2. The paper aims to protect data privacy but there is a lack of measurement of how privacy is protected. In all experiments, the paper focuses on showing how effectively the manipulated data degrades the test performance of the trained model. As far as I am concerned, this goal is the same as data poison attacks. However, it is unclear why and how the test performance relates to data privacy. From my understanding, the fact that a model performs badly on some samples doesn't necessarily imply the data privacy of those samples is protected. A thorough discussion on this issue will be necessary since protecting data privacy is the major goal of this work. Otherwise, it might be more appropriate to put it in the context of data poison attacks.

3. The paper claims that the unlearnability conferred by the noise generated via Eq. (4) is fragile to minor data transformation and proposes to adopt the expectation-over transformation technique (EOT) into the generation process. However, the paper doesn't present evidence to support this claim. Moreover, it is expected to include more ablation studies about the effect of the EOT technique in both the normal setting and the adversarial setting.

4. There is a lack of discussion of the data poison percentage that can effectively reduce the model performance.



**Summary Of The Paper:**

The paper proposes a min-min-max formulation to generate robust unlearnable examples in order to protect data privacy in adversarial learning. The basic idea of this paper builds upon a previous paper "Unlearnable examples: Making personal data unexploitable" where the error-minimization noise is generated to reduce the training loss such that the model performs badly on the clean data. This paper finds that such unlearnability is fragile in adversarial training and fragile to minor data augmentations. Therefore, it proposes to generate more robust unlearnable samples by considering adversarial loss and data augmentation in the noise generation process, and it shows the generated noise can effectively reduce the quality of the trained model through extensive empirical study.



**Summary Of The Review:**

Overall, the paper has shown that the proposed method can successfully reduce the test performance of the model (adversarially) trained by generating robust-error-minimization noise. However, the novelty is a bit limited and more justification of the data privacy protection is necessary. Therefore, I would like to recommend rejection.

---

> ### Author Response · Authors · 2021-11-14
> **To Reviewer ntwj**
>
> Thank you for your thorough review and constructive comments. All your concerns have been carefully responded below. The manuscript is carefully revised accordingly. We sincerely hope our responses fully address your questions.
>
>
> **Q1:** _The paper majorly builds upon the previous work on unlearnable examples which only consider normal training. Although this work improves the reliability of the noise generation by replacing the original loss with adversarial loss and considering the expectation of data augmentation, the proposed method is quite straightforward and a bit trivial. Therefore, the novelty of the paper is a bit limited._
>
> **A1:** We respectively argue that our work solves an important problem also pointed out in the "previous work" (Huang et al., 2021, Appendix D): protecting data from unauthorized exploitation by adversarial training. We agree that our method is natural and effective. It solves an important problem in a simple way.
>
>
> **Q2:** _The paper aims to protect data privacy but there is a lack of measurement of how privacy is protected. In all experiments, the paper focuses on showing how effectively the manipulated data degrades the test performance of the trained model. From my understanding, the fact that a model performs badly on some samples doesn't necessarily imply the data privacy of those samples is protected. A thorough discussion on this issue will be necessary since protecting data privacy is the major goal of this work._
>
> **A2:** This paper aims to prevent unauthorized exploitation of data, a new aspect of data privacy proposed by Huang et al. (2021) and different from the conventional understanding. In this setting, a significant decline of the test accuracy on the samples that are intended to protect is sufficient to indicate a competent algorithm. It shows that the learning algorithm does not learn much knowledge from the data; in other words, the data is well protected from unauthorized exploitation. We have added this discussion to Section 5.1 in the revised manuscript.
>
>
> **Q3:** _It is expected to include more ablation studies about the effect of the EOT technique in both the normal setting and the adversarial setting._
>
> **A3:** Thanks. We have added a detailed ablation study on EOT in the revised submission (in Section 5.2). A part of the results are shown in the following table:
>
> | Dataset | Adv. Train. Radius | REM with EOT | REM without EOT |
> | ---- | ---- | ---- | ---- |
> | CIFAR-10  | 0     | 27.09 | 18.55 |
> | CIFAR-10  | 4/255 | 48.16 | 89.41 |
> | CIFAR-100 | 0     | 10.14 |  7.03 |
> | CIFAR-100 | 4/255 | 27.10 | 63.79 |
>
> *Table: Test accuracy (%) of models trained on data protected by REM that is generated with or without the EOT technique. The adversarial training is conducted with perturbation radii $0$ and $4/255$, respectively. The defensive perturbation radius $\rho_u$ and the adversarial perturbation radius $\rho_a$ for REM are set as $8/255$ and $4/255$, respectively.*
>
> This table shows that without EOT, REM could not effectively protect data, which is in full agreement with our argument.
>
>
> **Q4:** _There is a lack of discussion of the data poison percentage that can effectively reduce the model performance._
>
> A4: Thanks. We have added a discussion about the impact of data poisoning percentage on the trained model performance in the revised submission (in Section 5.2) as below,
>
> >Table 2 shows that the performance of the trained model delines when the data protection percentage increases. This suggests that learning algorithms can learn less knowledge when more data is protected by REM, which fully support our algorithm.

---

> > ### Comment · Reviewer_ntwj · 2021-11-30
> > **New comments**
> >
> > Thanks for the detailed response. The ablation on EOT is very helpful.
> >
> > I can understand that the paper tries to prevent unauthorized exploitation of data, but I am not fully convinced that it is a way of protecting data privacy and such a claim might be misleading for the readers.
> >
> > In addition, there are several practical concerns:
> >
> > 1. Even when 100% of the data samples are perturbed, it requires the defensive epsilon in REM to be two times larger than the perturbation radius in adversarial training. In many adversarial ML papers, the perturbation budge is usually set as 8/255 or 16/255. Therefore, the proposed method requires a perturbation of 16/255 or 32/255, which seem to be large enough for detection.
> >
> > 2. I am doubtful about whether the proposed data poison is robust enough to degrade the performance. For instance, how does it performs when applying some very simple image preprocessing operation such as a low-pass filter?
> >
> > 3. In practice, it is almost impossible to perturb the whole dataset. Therefore, the results in Table 2 are important since it shows how the proposed method impacts the clean test performance when the ratio of perturbed data changes. However, I find the setting of Table 2 to be quite confusing, i.e., the setting of "Mixed" and "Clean" under different ratios. Moreover, could you please clarify the following concerns:
> >
> > (1) Why does the difference between the test accuracies on mixed data and clean data reflect the knowledge gained from the protected training data? Can you explain the settings clearly since I didn't find a description in the submission?
> >
> > (2) Why is there a significant performance gap between 80% and 100%? For instance, for ERM under 4/255, the performance drops from 82.98% to 30.69% (CIFAR10) and 63.05% to 27.10% (CIFAR100). Does it indicate that as long as there exist normal samples, the effectiveness of the proposed method becomes much weaker?

---

> > > ### Author Response · Authors · 2021-11-30
> > > **Responses to new comments (2/2)**
> > >
> > > **Q4.1:** _In practice, it is almost impossible to perturb the whole dataset. Therefore, the results in Table 2 are important since it shows how the proposed method impacts the clean test performance when the ratio of perturbed data changes. However, I find the setting of Table 2 to be quite confusing, i.e., the setting of "Mixed" and "Clean" under different ratios._
> > >
> > > **A4.1:** Thanks. Please let us clarify using the ratio "80\%" as an example: (1) "Mixed" means that the model is trained on the training dataset, where 80\% data has been processed by REM while the other 20\% is original data; and (2) "Clean" means that the model is trained on only 20\% original data. We will carefully revise our manuscript to make it clear.
> > >
> > > **Q4.2:** _Why does the difference between the test accuracies on mixed data and clean data reflect the knowledge gained from the protected training data? Can you explain the settings clearly since I didn't find a description in the submission?_
> > >
> > > **A4.2:** Thanks. Please let us still take the ratio "80\%" as an example. The difference between the two settings is the 80\% data that has been processed by REM; please also kindly refer to A4.1. Thus, the difference between the test accuracy on the "Mixed" setting and the "Clean" setting exactly indicates the knowledge learned from the additional 80\% data processed by REM. We will add a detailed discussion in the manuscript.
> > >
> > > **Q4.3:** _Why is there a significant performance gap between 80% and 100%? For instance, for ERM under 4/255, the performance drops from 82.98% to 30.69% (CIFAR10) and 63.05% to 27.10% (CIFAR100). Does it indicate that as long as there exist normal samples, the effectiveness of the proposed method becomes much weaker?_
> > >
> > > **A4.3:** Thanks. We respectfully note that this significant performance gap between 80\% and 100\% exactly demonstrates that adversarial learning can learn considerable knowledge from a small dataset without protection measures, while REM can protect data from unauthorized exploitation.
> > >
> > > In our manuscript, ResNet-18 is trained on 100\% data protected by REM, Mixed data (20\% of the original data and 80\% protected by REM), and 20\% original data. The test results demonstrate that considerable knowledge has been learned, as shown in the following table.
> > >
> > > | Dataset | 100\% clean dataset | Mixed (80\% protected by REM with $\rho_u=8$, 20\% clean dataset) | Clean (20\% clean dataset) | 100\% protected by REM with $\rho_u=8$ |
> > > | ---- | ---- | ---- | ---- | ---- |
> > > | CIFAR-10 | 92.37 | 82.98 | 83.37 | 30.69 |
> > > | CIFAR-100 | 68.91 | 52.33 | 47.99 | 14.15 |
> > >
> > > *Table: Test accuracy (%) of ResNet-18 trained with adversarial training perturbation budget $2/255$ on different datasets.*
> > >
> > > | Dataset | 100\% clean dataset | Mixed (80\% protected by REM with $\rho_u=16$, 20\% clean dataset) | Clean (20\% clean dataset) | 100\% protected by REM with $\rho_u=16$ |
> > > | ---- | ---- | ---- | ---- | ---- |
> > > | CIFAR-10 | 89.51 | 79.00 | 79.41 | 28.31 |
> > > | CIFAR-100 | 64.50 | 47.27 | 44.79 | 14.46 |
> > >
> > > *Table: Test accuracy (%) of ResNet-18 trained with adversarial training perturbation budget $4/255$ on different datasets.*
> > >
> > > We respectively note that this phenomenon exactly supports the effectiveness of REM. The central goal of REM is to protect data from unauthorized exploitation of adversarial learning. In other words, the goal is to make the protected data “useless” for attackers. The poor test accuracy on 100\% protected data shows that our REM has significant protection capabilities. The small gap (smaller than 5\%) between test accuracy of (80\% protected data by REM + 20\% original data) and 20\% original data demonstrates that scant knowledge is discovered from the 80\% protected data by REM.
> > >
> > > **References:**
> > >
> > > [1] Huang H., Ma X., Erfani S.M., Bailey J., Wang Y. "Unlearnable examples: Making personal data unexploitable". ICLR 2021.
> > > [2] Yuan C. H., Wu S. H. "Neural Tangent Generalization Attacks". ICML 2021.
> > > [3] Fowl L., Goldblum M., Chiang P., Geiping J., Czaja W., Goldstein T. "Adversarial examples make strong poisons". arXiv 2021.

---

> > > ### Author Response · Authors · 2021-11-30
> > > **Responses to new comments (1/2)**
> > >
> > > Thanks for your constructive comments. All your concerns have been responded below. The manuscript will be carefully revised accordingly in the next version due to time limitation. We sincerely hope all your concerns have been cleared.
> > >
> > > **Q1:** _I can understand that the paper tries to prevent unauthorized exploitation of data, but I am not fully convinced that it is a way of protecting data privacy and such a claim might be misleading for the readers._
> > >
> > > **A1:** Thanks. The word “privacy" will be replaced by "unauthorized exploitation of data" according to your suggestion in our manuscript to clear any ambiguity.
> > >
> > > We would also love to clarify that the following concept has been proposed and discussed in recent works [1-3]: "protecting user privacy by making scraped data useless for training models" [3]. We inherit the statement of “privacy” from them.
> > >
> > > **Q2:** _Even when 100% of the data samples are perturbed, it requires the defensive epsilon in REM to be two times larger than the perturbation radius in adversarial training. In many adversarial ML papers, the perturbation budge is usually set as 8/255 or 16/255. Therefore, the proposed method requires a perturbation of 16/255 or 32/255, which seem to be large enough for detection._
> > >
> > > **A2:** Thanks. We have conducted additional experiments according to your questions as show below. The results demonstrate that a large defensive epsilon would not make the images detectable for eyes, while a large adversarial learning perturbation budget significantly reduces the test accuracy on clean data.
> > >
> > > **Robust visualization of perturbed images in large-perturbation setting:** images are added robust error-minimizing noise with a perturbation of $16/255$ or $32/255$. Visualization demonstrates that the perturbations are still very slight for detection by human eyes. This visualization will be added to our manuscript.
> > >
> > > **Adversarial learning performance significantly declines in large-perturbation setting:** ResNet-18 is then trained on CIFAR-10 and CIFAR-100 with adversarial training perturbation budgets $8/255$ and $10/255$. The test accuracy on test data is presented below.
> > >
> > > | Dataset | $\rho_a$=0 | $\rho_a=8/255$ | $\rho_a=10/255$ |
> > > | ---- | ---- | ---- | ---- |
> > > | CIFAR-10 | 94.66 | 84.79 |82.24 |
> > > | CIFAR-100 | 76.27 | 58.27 | 55.57 |
> > >
> > > *Table: Test accuracy (%) of ResNet-18 trained with different adversarial training perturbation budgets $\rho_a$ on clean dataset.*
> > >
> > > This table shows that a large perturbation budge in adversarial training significantly undermines the test accuracy on clean data. When the adversarial training perturbation budget is $8/255$ (or $10/255$), the test accuracy is reduced by over 10\% (or 12\%) on CIFAR-10 and 18\% (or 21\%) on CIFAR-100. The performance decline of perturbation budget is $16/255$ is supposed to be even larger. These experiments will be added in our manuscript.
> > >
> > > **Q3:** _I am doubtful about whether the proposed data poison is robust enough to degrade the performance. For instance, how does it performs when applying some very simple image preprocessing operation such as a low-pass filter?_
> > >
> > > **A3:** Thanks. Given that the discussion period is about to end, we cannot manage to add more experimental results studying the defence performance of the robust error-minimizing noise against the low-pass filter at present. We will conduct additional experiments according to your suggestions, which will be added to the next version.

---

### Official Review · Reviewer_c7i6 · 2021-11-04

**Correctness:** 3
**Technical Novelty And Significance:** 3
**Empirical Novelty And Significance:** 2
**Recommendation:** 6
**Confidence:** 5

**Main Review:**

This paper is well-written, and the experiments are extensive. However, the proposed REM has the following issues:

* Unrealistic assumptions. The authors assume that a "data protector" knows the hyperparameters (such as the loss function (l_{1}, l_{2}, or l_{inf}) and perturbation budget (rho_{a})) of adversarial training to be used by a data consumer, which is unlikely to hold in practice. Although the author used a large perturbation budget to evaluate REM, the budget may differ across different applications for many reasons (e.g., due to different degradation in clean performance introduced by adversarial training). Also, recent studies (e.g., (Tramèr and Boneh 2019)) have demonstrated that adversarial training with a single type of perturbation cannot provide well defense against other types of adversarial attacks, yet adversarial training with multiple perturbation types is still an active research problem (Maini, Wong, and Kolter 2020; Madaan, Shin, and Hwang 2020; Zhang, Zhang, and Wang 2021; Stutz, Hein, and Schiele 2020). Furthermore, there is a recent study (Yuan and Wu, "Neural Tangent Generalization Attacks," 2021) that makes a dataset unlearnable using blackbox settings. The authors should 1) cite the work (Yuan et al. 2021) and discuss whether the problem targeted by this paper still exists for blackbox methods, and 2) discuss in more detail how REM perform against adversarial training with multiple perturbations, and 3) discuss how to choose the perturbation budget when running REM for different applications in practice.

* Time complexity. The REM finds unlearnable examples under adversarial training by approximating a min-min-max problem, which can be very time-consuming and limits the practicability of REM. How long does take to make a full MNIST, CIFAR10, CIFAR100, and ImageNet unlearnable, respectively?

Furthermore, exiting experiments lack some critical information and more should be conducted:

* None of an experiment shows how training and test accuracy evolve as the training epoch increases. There’s no evidence to prove that "the poor accuracy is not come from overfitting on the training data."

Typos:
1. Page 6 line 2: the loss has a typo
2. Table 7: "AP" should be "TAP"

Edit after rebuttal:

The authors have partially addressed my concerns, so I am willing to raise my score provided that the authors agree to further improve the paper in the following aspects:
1. Show the actual time cost of your approach and different baselines (including the new NTGA). Please describe the hardware (CPU/GPU) and settings you use in detail.
2. Discuss in more detail how to set the epsilon in practical situations, especially for the black-box settings.


**Summary Of The Paper:**

Based on a previous work (Huang et al. 2021), this paper proposed a method, called REM, that generates makes a dataset "unlearnable" to adversarial training. Experiments are conducted to validate the effectiveness of REM under various settings, including (1) different adversarial training perturbation radiuses, (2) different proportions of unlearnable data, (3) different model architectures, on different common datasets, including CIFAR-10, CIFAR-100, and ImageNet.

**Summary Of The Review:**

Overall, this paper studies an important and interesting problem. The paper is well-written and the experiments are extensive. However, the assumption used by the proposed REM is not realistic, and the authors missed an existing blackbox method that could entirely destroy the motivation of this work. Furthermore, some critical information, such as runtime, is missing from the current experiments. Therefore, I give a vote for rejection before the above concerns can be addressed.

---

> ### Author Response · Authors · 2021-11-14
> **To Reviewer c7i6 (2/2)**
>
> **Q3:** _None of an experiment shows how training and test accuracy evolve as the training epoch increases. There’s no evidence to prove that "the poor accuracy is not come from overfitting on the training data."_
>
> A3: Thanks. We have added experiment on how the training and test accuracy evolve in the revised manuscript. The results show that REM can progressively protect data from being learned along the adversarial training process. Please refer to Fig. 7 in Appendix C in the manuscript for details.
>
> **To typos:**
>
> **Q4:** _Page 6 line 2: the loss has a typo._
>
> **A4:** Thanks and addressed.
>
>
> **Q5:** _Table 7: "AP" should be "TAP"._
>
> **A5:** Thanks and addressed.

---

> ### Author Response · Authors · 2021-11-14
> **To Reviewer c7i6 (1/2)**
>
> Thank you for your thorough review and constructive comments. All your concerns have been carefully responded below. The manuscript is carefully revised accordingly. We sincerely hope our responses fully address your questions.
>
> **To major technical concerns:**
>
> **Q1.1:** _Recent studies (e.g., (Tramèr and Boneh 2019)) have demonstrated that adversarial training with a single type of perturbation cannot provide well defense against other types of adversarial attacks, yet adversarial training with multiple perturbation types is still an active research problem (Maini, Wong, and Kolter 2020; Madaan, Shin, and Hwang 2020; Zhang, Zhang, and Wang 2021; Stutz, Hein, and Schiele 2020). The authors should discuss in more detail how REM perform against adversarial training with multiple perturbations._
>
> A1.1: Thanks. Our method can well defend multiple types of adversarial training even when the perturbation norm is agnostic. We conducted additional experiments to verify this: (1) images are manipulated by REM with perturbation norm $L_\infty$-norm; and (2) adversarial training with perturbation norms $L_1$-norm and $L_2$-norm are applied to exploit the manipulated images. The results are collected below:
>
> | Dataset | Data Type | $L_1$-norm, Adv. Train. Radius $\rho_a = 200/255$ | $L_2$-norm, Adv. Train. Radius $\rho_a = 100/255$ |
> | ---- | ---- | ---- | ---- |
> | CIFAR-10  | Clean | 94.47 | 89.91 |
> | CIFAR-10  | REM   | 27.69 | 62.76 |
> | CIFAR-100 | Clean | 75.85 | 65.65 |
> | CIFAR-100 | REM   | 11.29 | 47.16 |
>
> *Table 1: Test accuracy (%) of models trained with different types of adversarial perturbations. The defensive perturbation radius $\rho_u$ and the adversarial perturbation radius $\rho_a$ of the robust error-minimizing noise (REM) are set as $8/255$ and $4/255$, respectively.*
>
> This table supports that REM is agnostic to perturbation norm. More experiments will be presented in the revised version.
>
> **Q1.2:** _Although the author used a large perturbation budget to evaluate REM, the budget may differ across different applications for many reasons (e.g., due to different degradation in clean performance introduced by adversarial training). The authors should discuss how to choose the perturbation budget when running REM for different applications in practice._
>
> **A1.2:** Our experiments show that a sufficiently large defense budget guarantees REM can well protect data from unauthorized exploitation, as long as the defensive epsilon in REM is two times larger than the perturbation radius in adversarial training. Hence, the perturbation budget is recommended to be sufficiently large as long as the manipulated images are still of high quality.
>
>
> **Q1.3:** _There is a recent study (Yuan and Wu, "Neural Tangent Generalization Attacks," 2021) that makes a dataset unlearnable using black-box settings. The authors should cite the work (Yuan and Wu 2021) and discuss whether the problem targeted by this paper still exists for black-box methods._
>
> **A1.3:** Thanks. We have cited and carefully discussed Yuan et al. (2021) in the revised version. We also conducted additional experiments to investigate whether Yuan et al. (2021) can protect data from exploitation by adversarial training, based on the manipulated CIFAR-10 dataset (named “CIFAR-10 - CNN(best)”) released by the authors (see [https://github.com/lionelmessi6410/ntga](https://github.com/lionelmessi6410/ntga)). Our experiment results show that Yuan et al. (2021) does not protect well data from adversarial training:
>
> | Adv. Train. Radius $\rho_a$ | 0/255 | 2/255 | 4/255 |
> | ---- | ---- | ---- | ---- |
> | Test Accuracy (%) | 11.89 | 75.07 | 86.91 |
>
> *Table2: Test accuracy (%) of models trained on CIFAR-10 that are protected by Yuan et al. (2021). The protected dataset is downloaded from the official repository of Yuan et al. (2021).*
>
> Table 2 shows that adversarial training achieves significantly high accuracy, which suggests that adversarial training can still learn knowledge from data manipulated by Yuan et al. (2021).
>
>
>
> **Q2:** _The REM finds unlearnable examples under adversarial training by approximating a min-min-max problem, which can be very time-consuming and limits the practicability of REM. How long does take to make a full MNIST, CIFAR10, CIFAR100, and ImageNet unlearnable, respectively?_
>
> **A2:** Thanks for this important question. The time cost for making CIFAR-10, CIFAR-100, and ImageNet unlearnable are presented as follows,
>
> | CIFAR-10 | CIFAR-100 | ImageNet (Subset) |
> | ---- | ---- | ---- |
> | 22.6 h | 22.6 h | 51.2 h |
>
> The time cost is acceptable to us. REM is supposed to process personal data to be unlearnable before releasing. The time cost above shows REM is competent.

---

> > ### Comment · Reviewer_c7i6 · 2021-11-18
> > **Low-frequency pertubations**
> >
> > The reviewer thanks the authors for their response. How do the lower-frequency poisoning in "FNN (1)" and "CNN (1)" from Yuan et al. (2021) work under adversarial training? Does the problem still exist?

---

> > > ### Author Response · Authors · 2021-11-19
> > > **Low-frequency poisoning cannot protect data from adversarial training**
> > >
> > > Thank you very much for your reply! We have conducted experiments to investigate whether lower-frequency poisoning in "FNN (1)" and "CNN (1)" from Yuan et al. (2021) can protect data from exploitation from adversarial training. The empirical results suggest that lower-frequency poisoning cannot defend against adversarial training:
> > >
> > > | NTGA Type | ERM ($\rho_a=0$) | Adversarial Training ($\rho_a=2/255$) | Adversarial Training ($\rho_a=4/255$) |
> > > | --------- | ------------------------------- | ------------------------------- | ------------------------------- |
> > > | CNN (1)    | 13.37                           | 75.04                           | 86.56                           |
> > > | FNN (1)    | 17.65                           | 80.94                           | 86.98                           |
> > > | FNN (best) | 21.25                           | 86.15                           | 86.11                           |
> > >
> > > *Table: Test accuracy (%) of models trained on CIFAR-10 that are protected by Yuan et al. (2021). The protected dataset is downloaded from the official repository of Yuan et al. (2021).*
> > >
> > > This table shows that the adversarial training achieves considerably high test accuracy even when the data is protected by low-frequency poisoning.

---

> > ### Comment · Reviewer_c7i6 · 2021-11-22
> > **Updated score if further improvements can be made**
> >
> > The authors have partially addressed my concerns, so I am willing to raise my score provided that the authors agree to further improve the paper in the following aspects:
> > 1. Show the actual time cost of your approach and different baselines (including the new NTGA). Please describe the hardware (CPU/GPU) and settings you use in detail.
> > 2. Discuss in more detail how to set the epsilon in practical situations, especially for the black-box settings.
> > 3. Discuss how the amount of poisoning in training set and the epsilon used by adversarial training could interact with each other. For example, what happens if the poisoning applies only to a part of an input image and the amount of poisoning is larger than the epsilon?

---

> > > ### Author Response · Authors · 2021-11-22
> > > **New experiments and discussions (3/3)**
> > >
> > > * **Model Training**
> > >
> > >   We follow Eq. (2) to perform adversarial training (Madry et al., 2018). Similar to that in training noise generator, we also focus on $L_\infty$-bounded noise $\|\rho_a\|_\infty \leq \rho_a$ in adversarial training.
> > >
> > >   In every experiment, the model is trained with SGD for $40,000$ iterations, with a batch size of $128$, a momentum factor of $0.9$, a weight decay factor of $0.0005$, an initial learning rate of $0.1$, and a learning rate scheduler that decays the learning rate by a factor of $0.1$ every $16,000$ iterations. For CIFAR-10 and CIFAR-100, the steps number $K_a$ and the step size $\alpha_a$ in PGD are set as $10$ and $\rho_a/5$. For the ImageNet subset, the steps number $K_a$ and the step size $\alpha_a$ are set as $8$ and $\rho_a/4$.
> > >
> > >
> > > **Q2:** *Discuss in more detail how to set the epsilon in practical situations, especially for the black-box settings.*
> > >
> > > **A2:** Thanks. We will add thorough empirical analysis and discussions in the next version.
> > >
> > > As a preliminary recommendation, practioners are recommended to set the defensive epsilon $\rho_u$ as large as possible in the black-box settings, while the manipulated images are still of sufficiently high quality. In the current experiments, our algorithm has fairly good defense performance when the defensive epsilon $\rho_u$ is larger than two times the adversarial epsilon $\rho_a$. We have added a visualization as Fig. 8 in Appendix C to illustrate our recommendation.
> > >
> > >
> > > **Q3.1:** *Discuss how the amount of poisoning in training set and the epsilon used by adversarial training could interact with each other. For example, what happens if the poisoning applies only to a part of an input image and the amount of poisoning is larger than the epsilon?*
> > >
> > > **A3.1:** Thanks. We have added a detailed discussion in Section 5.2 according to your suggestion as below.
> > >
> > > > Table 2 shows that when the adversarial training perturbation radius is small, robust error-minimizing noise can effectively protect the selected part of the data. However, when a large adversarial training perturbation radius presents, the protection becomes worthless. This suggests that to protect data privacy against adversarial training with a perturbation radius $\rho_a$, one has to set the defensive perturbation radius $\rho_u$ of the robust error-minimizing noise to a value that is relatively larger than $\rho_a$. Table 2 shows that as the data protection percentage decreases, the performance of the trained model increases. This suggests that the model can learn more knowledge from more clean data, which coincides with intuition. Nevertheless, Table 2 further shows that the privacy preservation ability of the robust error-minimizing noise is stronger than that of the error-minimizing noise and the targeted adversarial poisoning noise. This demonstrates that the robust error-minimizing noise is still more favorable than other types of defensive noise when adversarial training presents.
> > >
> > > **Q3.2:** *What happens if the poisoning applies only to a part of an input image and the amount of poisoning is larger than the epsilon?*
> > >
> > > **A3.2:** Thanks. We will conduct experiments to study what will happen if only a part of an input image is poisoned and the amount of poisoning is larger than the epsilon.

---

> > > ### Author Response · Authors · 2021-11-22
> > > **New experiments and discussions (2/3)**
> > >
> > > **Experiment settings:** we have presented detailed experiment settings in Appendix A.3 and Appendix A.4. Parts of them are shown below.
> > >
> > > * **Defensive Noise Generation**
> > >
> > >   Our experiments involve three types of defensive noise, the proposed robust error-minimizing noise, and two baseline methods, error-minimizing noise and adversarial poisoning noise.
> > >
> > >   | Datasets             | Noise Type | $\alpha_u$   | $K_u$ | $\alpha_a$ | $K_a$ |
> > > | -------------------- | ---------- | ------------ | ----- | ---------- | ----- |
> > > | CIFAR-10 / CIFAR-100 | EM         | $\rho_u/5$   | $10$  | -          | -     |
> > > | CIFAR-10 / CIFAR-100 | TAP        | $\rho_u/125$ | $250$ | -          | -     |
> > > | CIFAR-10 / CIFAR-100 | REM        | $\rho_u/5$   | $10$  | $\rho_a/5$ | $10$  |
> > > | ImageNet (Subset)    | EM         | $\rho_u/5$   | $7$   | -          | -     |
> > > | ImageNet (Subset)    | TAP        | $\rho_u/50$  | $100$ | -          | -     |
> > > | ImageNet (Subset)    | REM        | $\rho_u/4$   | $7$   | $\rho_a/5$ | $10$  |
> > >
> > >   *Table 5: The settings of PGD (see Eq. (3)) for the noise generations of error-minimizing noise (EM), targeted adversarial poisoning noise (TAP), and robust error-minimizing noise (REM) in different experiments. $\rho_u$ denotes the defensive perturbation radius of different types of noise, while $\rho_a$ denotes the adversarial perturbation radius of the robust error-minimizing noise.*
> > >
> > > * **Robust Error-minimizing Noise**
> > >
> > >   Following Huang et al. (2021), we employ ResNet-18 (He et al., 2016) as the source model $f'$ for the robust error-minimizing noise generation.
> > >
> > >   The $L_\infty$-bounded noises $\|\delta_u\|_\infty \leq \rho_u$ and $\|\delta_a\|_\infty \leq \rho_a$ are adopted in our experiments, in which the defensive perturbation radius $\rho_u$ and adversarial perturbation radius $\rho_a$ can take various values. The settings of PGD for solving the inner minimization and maximization problems in Eq. (5) are presented in Table 5.
> > >
> > >   For CIFAR-10 and CIFAR-100, each source model is trained with SGD for $5,000$ iterations, with a batch size of $128$, a momentum factor of $0.9$, a weight decay factor of $0.0005$, an initial learning rate of $0.1$, and a learning rate scheduler that decay the learning rate by a factor of $0.1$ every $2,000$ iterations. For EOT, the data transformation $T$ is set as the data augmentation of the corresponding dataset, and the repeatedly sampling number for expectation estimation is set as $5$.
> > >
> > >   Besides, for the ImageNet subset, each source model is trained with SGD via Eq. (5) for $3,000$ iterations, with a batch size of $128$, a momentum factor of $0.9$, a weight decay factor of $0.0005$, an initial learning rate of $0.1$, and a learning rate scheduler that decay the learning rate by a factor of $0.1$ every $1,200$ iterations. For EOT, the data transformation $T$ is set as the data augmentation of the corresponding dataset, and the repeatedly sampling number for expectation estimation is set as $4$.
> > >
> > > * **Implementations of Baseline Methods**
> > >
> > >   Two baseline methods are adopted in our experiments as comparisons, including the error-minimizing noise method and the targeted adversarial poisoning noise method. Every method is reproduced on our own.
> > >
> > > 	* **Error-minimizing noise (Huang et al., 2021).** We follow Eq. (3) to train the error-minimizing noise generator. The error-minimizing noise is then generated with the trained noise generator. ResNet-18 model is used as the source model $f'$. PGD is employed for solving the inner minimization problem in Eq. (3), where the settings of PGD is presented in Table 5. Other hyperparameters for training the noise generator are set the same as that for the robust error-minimizing noise generator in the previous section.
> > >
> > > 	* **Targeted adversarial poisoning noise (Fowl et al., 2021b).** This type of noise is generated via conducting targeted adversarial attack to the model that is trained on clean data, in which the generated adversarial perturbation is used as the adversarial poisoning noise. Specifically, given a fixed model $f_0$ and a sample $(x,y)$, the targeted adversarial attack will generate noise via solving the problem $\arg\max_{\|\delta_u\| \leq \rho_u} \ell(f_0(x + \delta_u), g(y))$, where $g$ is a permutation function on the label space $\mathcal{Y}$. PGD with *differentiable data augmentation* (Geiping et al., 2021) is employed for solving the above problem. The hyper-parameters for the PGD is given in Table 5.

---

> > > ### Author Response · Authors · 2021-11-22
> > > **New experiments and discussions (1/3)**
> > >
> > > Thank you very much for your precious suggestions and kind support! We are working on additional experiments according to your suggestions. Some of the experiments have been ready. The results are in full agreement with our claims, as discussed below. All the experiments and discussions will be presented in the manuscript.
> > >
> > > **Q1:** *Show the actual time cost of your approach and different baselines (including the new NTGA). Please describe the hardware (CPU/GPU) and settings you use in detail.*
> > >
> > > **A1:** Thanks. The actual time cost of our approach and other baselines (EM and TAP) are presented in the following table:
> > >
> > > | Dataset           | EM    | TAP   | REM    |
> > > | ----------------- | ----- | ----- | ------ |
> > > | CIFAR-10          | 0.4 h | 0.5 h | 22.6 h |
> > > | CIFAR-100         | 0.4 h | 0.5 h | 22.6 h |
> > > | ImageNet (subset) | 3.9 h | 5.2 h | 51.2 h |
> > >
> > > *Table: Time cost for making the whole dataset unlearnable with different methods.*
> > >
> > > **NTGA:** we will add NTGA as a new baseline in the next version.
> > >
> > > **Hardware:** (1) the experiments on CIFAR-10 and CIFAR-100 are conducted on 1 GPU (NVIDIA Tesla V100 16GB) and 10 CPU cores (Intel Xeon CPU E5-2650 v4 @ 2.20GHz); and (2) the experiments on ImageNet are conducted on 4 GPU (NVIDIA Tesla V100 16GB) and 40 CPU cores (Intel Xeon CPU E5-2650 v4 @ 2.20GHz).

---

### Public Comment · ~Xingjun_Ma1 · 2021-11-09
**Interesting Work**

It is interesting to see how the limitation against adv train can be largely solved by the proposed min-min-max framework. It surprises me that the two sets of noises (adv and EM) are compatible with each other. I wonder if the authors could share more insights on patterns found by REM in Fig. 3? It looks like aligns well with the robust features (shapes) in the images. Is that the reason why adv train can be bypassed?

---

> ### Author Response · Authors · 2021-11-14
> **Insight of alignment between REM noises and robust shapes**
>
> Thank you very much for your attention to our work! It is indeed an interesting observation that the patterns found by REM align well with the robust features/shapes. This phenomenon robustly exists when the images are of sufficiently high-resolution, as shown in the experiments conducted on ImageNet; please kindly see Appendix B. Our insight/hypothesis is that REM can identify robust features in a data-driven way and then manipulate this part. In this way, REM has advantages in protecting data against unauthorized exploitation by adversarial training.

---

### Decision · Program_Chairs · 2022-01-20

**Decision:**

Accept (Poster)

**Comment:**

To address the problem of unauthorized use of data, methods are proposed to make data unlearnable for deep learning models by adding a type of error-minimizing noise. Based on th fact that the conferred unlearnability is found fragile to adversarial training, the authors design new methods to generate robust unlearnable examples that are protected from adversarial training. In addition, considering the vulnerability of error-minimizing noise in adversarial training, robust error-minimizing noise is then introduced to reduce the adversarial training loss.
The authors have tried to respond to reviewers' comments along with adding more experiments.
Overall, this manuscript finally gets three positive reviews and one negative review, where the possible vulnerability or robustness of error-minimizing noise against (simple) image processing operations was not verified.
In comparison with other manuscripts I'm handling that got consistent positive comments, this manuscript is still recommended to be accepted (poster) with a further study of robustness under simple image transformations in the final version.